# Nuclear receptor E75/NR1D2 promotes tumor malignant transformation by integrating Hippo and Notch pathways

Xianping Wang [1,2,6✉], Yifan Guo[1,2,6], Peng Lin [1,2,6], Min Yu[3], Sha Song[1,2], Wenyan Xu [1,2], Du Kong [1,2], Yin Wang [1,2,5], Yanxiao Zhang [1,2], Fei Lu[4], Qi Xie [1,2✉] & Xianjue Ma [1,2✉]

## Abstract

Hormone therapy resistance and the ensuing aggressive tumor progression present a significant clinical challenge. However, the mechanisms underlying the induction of tumor malignancy upon inhibition of steroid hormone signaling remain poorly understood. Here, we demonstrate that *Drosophila* malignant epithelial tumors show a similar reduction in ecdysone signaling, the main steroid hormone pathway. Our analysis of ecdysone-induced downstream targets reveals that overexpression of the nuclear receptor E75, particularly facilitates the malignant transformation of benign tumors. Genome-wide DNA binding profiles and biochemistry data reveal that E75 not only binds to the transcription factors of both Hippo and Notch pathways, but also exhibits widespread co-binding to their target genes, thus contributing to tumor malignancy. We further validated these findings by demonstrating that depletion of *NR1D2*, the mammalian homolog of E75, inhibits the activation of Hippo and Notch target genes, impeding glioblastoma progression. Together, our study unveils a novel mechanism by which hormone inhibition promotes tumor malignancy, and describes an evolutionarily conserved role of the oncogene E75/NR1D2 in integration of Hippo and Notch pathway activity during tumor progression.

**Keywords** Steroid Hormone; Hippo Signaling; Notch Signaling; NR1D2; Glioblastoma
**Subject Categories** Cancer; Chromatin, Transcription & Genomics; Signal Transduction

## Introduction

Hormones play a pivotal role in controlling cell division and growth, and their dysregulation can lead to uncontrolled cell proliferation and tumor formation (Conway-Campbell et al, 2007;

Metcalfe et al, 2018; Xu et al, 2021). Many hormones can function as growth factors, thereby stimulating cell proliferation. For instance, estrogen has been shown to promote the growth of hormone receptor-positive breast cancer cells, and women who had menopausal hormone therapy shortly after menopause have been found to have a significantly increased risk of developing invasive breast cancer (Kotsopoulos, 2019). Similarly, the male sex hormone androgens can stimulate prostate cancer cells to grow through the activation of androgen receptors. Consequently, hormone therapy is widely employed as a treatment strategy for both breast and prostate cancers, yielding highly effective initial responses in terms of impeding tumor growth (Asemota et al, 2022; Caffa et al, 2020; Metcalfe et al, 2018). However, it is important to note that the development of resistance to hormone therapy poses a significant clinical problem (Hanker et al, 2020; Metcalfe et al, 2018). In the case of breast cancer, around 30–40% of patients experience this resistance (Anurag et al, 2018). More strikingly, most male patients undergoing androgen deprivation therapy eventually progress to advanced castration-resistant prostate cancer (CRPC) (Buttigliero et al, 2015). The resistant tumor cells have increased proliferation ability and become more aggressive, and patients have a worse survival outcome (Brankovic-Magic et al, 2002; Rani et al, 2019). Nonetheless, the underlying mechanisms that trigger the malignant transformation in hormone-signaling-reduced tumor cells remain largely unexplored.

*Drosophila melanogaster* is a widely used model organism in cancer research due to its genetic tractability and the high degree of conservation of key cancer-related pathways (Bilder et al, 2021; Liu et al, 2022b; Verheyen, 2022; Villegas, 2019). The evolutionarily conserved Hippo and Notch signaling pathways, both initially discovered and characterized in *Drosophila*, play crucial roles in cell proliferation, differentiation, and apoptosis (Zheng and Pan, 2019; Zhou et al, 2022). The activation of the transcription complexes of Hippo and Notch pathways, namely the YAP/TAZ-TEAD and NICD-RBPJ-MAML complexes, respectively, promotes tumor progression and malignancy, including glioblastoma (GBM), a highly aggressive type of brain cancer (Orr et al, 2011; Purow et al, 2005). Previous studies have uncovered a complex interplay

[1]Westlake Laboratory of Life Sciences and Biomedicine, Hangzhou 310024 Zhejiang, China. [2]School of Life Sciences, Westlake University, Hangzhou 310024 Zhejiang, China. [3]Institute of Biomedicine and Biotechnology, Shenzhen Institute of Advanced Technology, Chinese Academy of Sciences, Shenzhen, China. [4]State Key Laboratory of Chemical Oncogenomics, Key Laboratory of Chemical Genomics, Peking University Shenzhen Graduate School, Shenzhen 518055, China. [5]Present address: Department of Diabetes & Cancer Metabolism, Beckman Research Institute of City of Hope National Medical Center, Duarte, CA 91010, USA. [6]These authors contributed equally: Xianping Wang, Yifan Guo, Peng Lin. ✉E-mail: wangxianping@westlake.edu.cn; xieqi@westlake.edu.cn; maxianjue@westlake.edu.cn

between these two pathways. For instance, activation of YAP/TAZ can increase the expression of Notch receptors and ligands, thereby activating Notch signaling (Slemmons et al, 2017). Conversely, Notch signaling can inhibit the Hippo pathway to increase the activity of YAP/TAZ, thus promoting cell proliferation and inhibiting apoptosis (Kim et al, 2017). However, the precise molecular mechanisms underlying the integration of the Hippo and Notch pathways during cancer pathogenesis remain incompletely understood.

The hormonal system of *Drosophila*, while less complex than that of mammals, exhibits several similarities to its mammalian counterparts. As a result, *Drosophila* serves as a valuable tool in the field of hormonal research (Verheyen, 2022). The investigation of ecdysone, a primary steroid hormone in *Drosophila*, has shed light on its diverse roles in both physiological and pathological circumstances, ranging from the regulation of intestinal stem cell fate and morphogenesis to collective cell migration and tumor progression (Ahmed et al, 2020; Jiang et al, 2018; Santabárbara-Ruiz and Léopold, 2021; Wang et al, 2020; Zipper et al, 2020). Ecdysone binds to and activates ecdysone receptor (EcR), which in turn binds to specific regions of the DNA known as ecdysone response elements (EcREs), thus initiating a cascade of ecdysone-responsive transcription factor expression, including *Eip74EF*, *Eip75B*, *Eip93*, *Br*, *Hr3*, and *ftz-f1*, among others (Huet et al, 1995; Sullivan and Thummel, 2003; White et al, 1997). The early ecdysone-responsive gene *Eip75B* (*E75*) encodes a heme-binding nuclear receptor and has been implicated in various biological processes, including cell growth and differentiation (Ahmed et al, 2020; Reinking et al, 2005; Zipper et al, 2020), whereas the roles and mechanisms of *Eip75B* in tumorigenesis remain unknown.

In this study, we compared the transcriptome difference between benign and malignant *Drosophila* epithelium tumors and discovered that ecdysone steroid hormone signaling is specifically inhibited in malignant tumors. Furthermore, we demonstrated that ectopic expression of *E75* is sufficient to transform benign tumors into malignant ones. E75 drives cell proliferation and tumor malignancy by integrating Hippo and Notch signaling pathways at the transcription factor level. Additionally, we showed that *NR1D2*, the mammalian ortholog of *E75*, has a functionally and mechanically conserved role in regulating GBM progression in vivo.

# Results

## Ecdysone steroid hormone signaling is inhibited in *Drosophila* epithelial malignant tumor

*Drosophila* tumors can be classified into two main subtypes: neoplastic and hyperplastic (Bilder et al, 2021). Neoplastic tumors exhibit signs of cell polarity loss and differentiation defects, resulting in the formation of multi-layered invasive tumors. The typical genes of this category include *scribble* (*scrib*), *discs large* (*dlg*), and *lethal giant larvae* (Greenman et al, 2007) (Bilder et al, 2000; Bilder et al, 2021). Conversely, hyperplastic tumors maintain the cell architecture of the single-layered epithelium and preserve the cell fate. Examples of these tumor-regulating genes include *warts/lats* (*wts*) and oncogenic *Ras* (*Ras^{V12}*). We generated GFP-labeled clones harboring mutations of different tumor suppressor genes and oncogenes using the mosaic analysis with a repressible

cell marker (MARCM) system (Fig. 1A) (Brumby and Richardson, 2003; Lee and Luo, 2001). In the presence of surrounding wild-type (WT) cells, the *scrib* mutant clones undergo elimination through a conserved process known as tumor-suppressive cell competition (Figs. 1A,B and EV1A) (Brumby and Richardson, 2003; Katsukawa et al, 2018; Kong et al, 2022; Sanaki et al, 2020; Zheng et al, 2023). However, when *Ras^{V12}* is co-expressed, the *scrib* clones are transformed into malignant tumors, and tumor-bearing animals become giant larvae on the 9th day (Fig. 1B,B') (Brumby and Richardson, 2003; Liu et al, 2022a; Pagliarini and Xu, 2003). Interestingly, while mutation of *wts* alone resulted in larger clones, co-deletion of *wts* failed to transform *scrib* clones into large tumor masses (Figs. 1B,B' and EV1A).

To investigate the mechanisms underlying the phenotypic differences between the *scrib^{−/−},wts^{−/−}* and *scrib^{−/−},Ras^{V12}* tumors (Fig. 1B'), we conducted bulk RNA-seq analysis. Interestingly, by analyzing the differentially expressed genes (DEGs), we noticed significant enrichment of multiple hormone-related terms (Figs. 1C and EV1B). We further performed motif enrichment analysis using HOMER (Hypergeometric Optimization of Motif EnRichment) (Heinz et al, 2010) to identify the crucial transcription factors (TFs) involved in the regulation of the DEGs. Of note, the highest-ranked TF identified was *broad* (*br*) (Fig. 1D), a known "early" ecdysone-induced gene that subsequently triggers the activation of "late" genes in the ecdysone signaling. Moreover, the gene set enrichment analysis (GSEA) and RNA-seq analysis both demonstrated a significant down-regulation of the ecdysone signaling in the *scrib^{−/−},Ras^{V12}* malignant tumor compared to the *scrib^{−/−},wts^{−/−}* benign tumors (Figs. 1E and EV1C). Consistent with these findings, the transcriptional activation of ecdysone signaling in vivo was severely impeded in malignant tumors, but only mildly inhibited in benign tumors, as demonstrated by the ecdysone response element (*EcRE*)-driven LacZ (*EcRE-LacZ*) reporter and Br staining (Fig. 1F–G'). Another malignant tumor induced by *lgl^{−/−}*, *Ras^{V12}* also exhibited a significant reduction in Br (Fig. EV1D,D'). In line with the potential tumor-suppressive role of increased ecdysone signaling, we observed a robust suppression of *lgl^{−/−}*, *Ras^{V12}*-induced tumorigenesis and restoration of the pupation defect upon hyperactivation of ecdysone signaling through ectopic expression of three different isoforms of *EcR* (*EcRA*, *EcRB1*, and *EcRC*) (Fig. EV1E–E''), accompanied with restoration of Br (Fig. EV1H,H'). It's worth to noting that ectopic expression of three different isoforms of *EcR* alone did not affect clonal growth (Fig. EV1F,F'). Collectively, these findings indicate that the inhibition of ecdysone signaling might play a critical role in the malignant transformation of tumors.

## E75 overexpression induces tumor malignant transformation

To further investigate the potential role of EcR signaling in regulating tumor progression, we overexpressed several well-established downstream target genes of EcR in the *scrib^{−/−}*, *wts^{−/−}* tumors, including *Hr3*, *ftz-f1*, *Eip93F*, *br*, and *E75*. Surprisingly, we observed that only the overexpression of *E75* significantly enhanced the overgrowth of *scrib^{−/−},wts^{−/−}* tumors, while the expression of other genes impeded tumorigenesis (Fig. 1H,H'). These findings demonstrate the complexity of EcR target genes in the regulation of tumorigenesis and highlight the distinctive role of E75 in driving

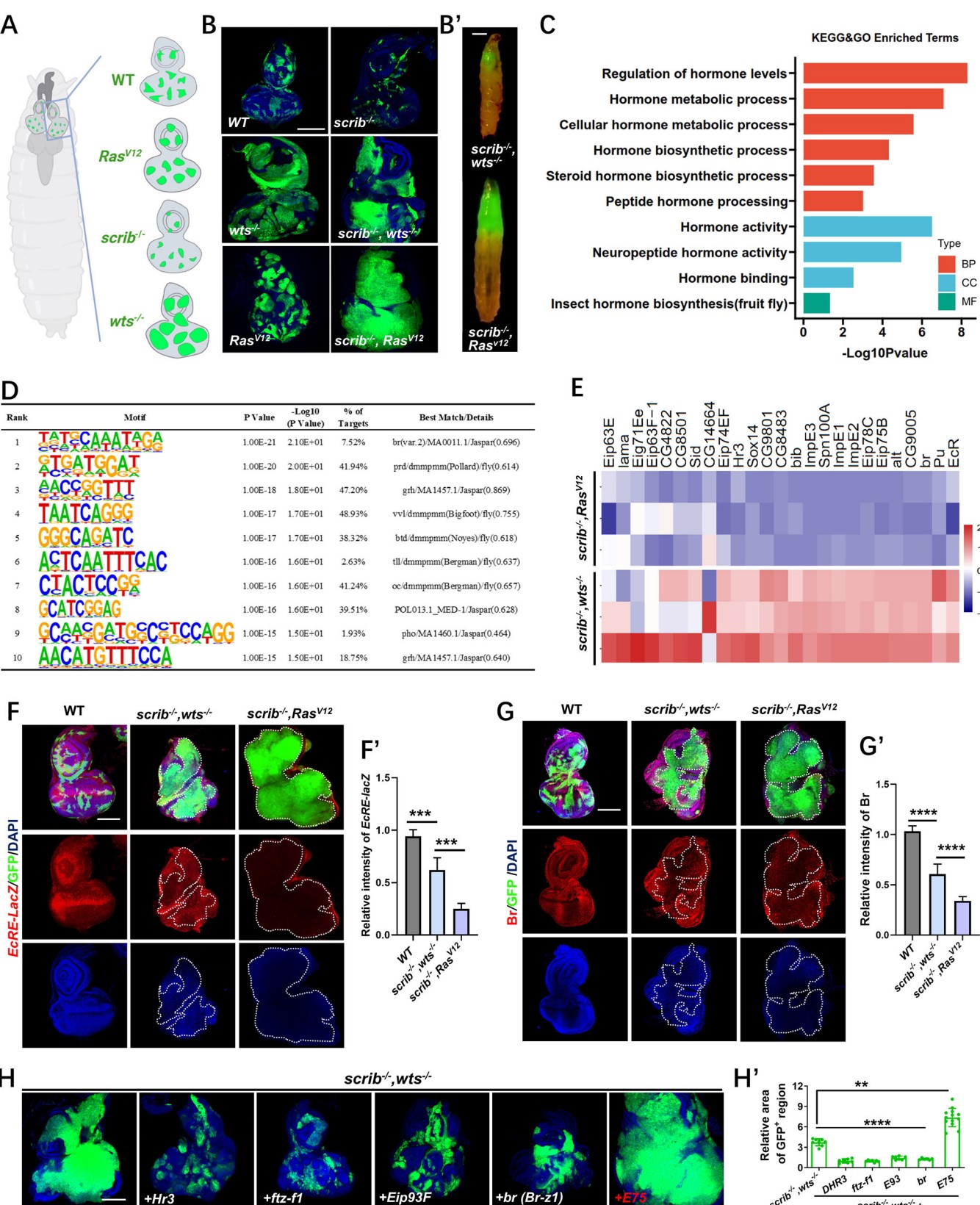

**Figure 1. Ecdysone steroid hormone signaling is inhibited in *Drosophila* epithelial malignant tumor.**

(A) Schematic representation of eye-antennal discs from a *Drosophila* larva of *ey-Flp*-MARCM-induced mosaics. (B) Eye-antennal discs bearing *ey-Flp*-MARCM-induced mosaics of indicated genotypes. (B') Representative images of tumor-bearing larvae of *scrib⁻/⁻,wts⁻/⁻* and *scrib⁻/⁻,Ras^{V12}* on the 6th day and 9th day, respectively. (C) Enrichment of hormone-related terms between *scrib⁻/⁻,wts⁻/⁻* and *scrib⁻/⁻,Ras^{V12}* tumors from KEGG and GO analyses. The hypergeometric distribution examination $p < 0.1$ served as the filtration threshold. Here, we displayed the terms associated with 'hormone' with their $-\log_{10}$ transformed $p$ value. (D) HOMER motif analysis of ecdysone-related genes in *scrib⁻/⁻,wts⁻/⁻* and *scrib⁻/⁻,Ras^{V12}* tumors. ZOOPS scoring and hypergeometric distribution test were considered collectively, and motifs with $p < 0.0000001$ were considered significantly enriched. (E) Heatmap profiles of ecdysone signaling-related genes in *scrib⁻/⁻,wts⁻/⁻* and *scrib⁻/⁻,Ras^{V12}* tumors. (F, G) Confocal images of eye-antennal discs bearing *ey-Flp*-MARCM-induced mosaics of indicated genotype stained with anti-β-galactosidase antibody for the *EcRE-LacZ* staining (F) or anti-Br antibody (G). Quantification of the relative intensity of *EcRE-LacZ* (F', from left to right, $n = 9, 8, 8$) and Br (G', $n = 5, 8, 11$) in GFP positive mosaics clones. $n$ represents the number of biological replicates. Statistical analysis by Ordinary one-way ANOVA test; mean ± SD. $p$ values (F'), from left to right, ***$p = 0.0005$, ***$p = 0.0002$. $p$ values (G'), ****$p = 0.000000916$, ****$p = 0.000000045$. (H) Confocal images of eye-antennal discs bearing *ey-Flp*-MARCM-induced tumors and mosaics of indicated genotypes. (H') Quantification of the relative size of GFP positive regions ($n = 8, 7, 7, 6, 13$). $n$ represents the number of biological replicates. Statistical analysis by ordinary one-way ANOVA test; mean ± SD. **$p = 0.0019$, ****$p = 0.0000013$, ****$p = 0.00000037$, ****$p = 0.0000076$, ****$p = 0.00000079$. Scale bars: 100 µm (B, F, G, H), 500 µm (B'). Source data are available online for this figure.

tumor overgrowth (Fig. EV1K). Previous studies have shown that E75 genetically represses the ecdysone-triggered cascade (Hiruma and Riddiford, 2004; Johnston et al, 2011). In line with this, *ey-Flp*-MARCM-induced clonal overexpression of the A isoform of *Eip75B* (abbreviated as *E75*) significantly inhibited the endogenous ecdysone signaling activation (Fig. EV1I,J). Moreover, the co-expression of EcR partially repressed the growth phenotype caused by E75 overexpression (Fig. EV1G,G'). Together, these findings imply that E75-induced overgrowth is partially reliant on the inhibition of EcR.

Differentiation failure is one of the key characteristics that distinguish malignant from benign tumors. Clones overexpressing *E75* exhibited a dramatic defect in differentiation, as evidenced by the staining of the neuronal differentiation marker Elav (Fig. 2A). Moreover, overexpression of *E75* exhibited a synergistic effect when combined with either *scrib⁻/⁻* or *wts⁻/⁻*, resulting in tumor overgrowth and pupation defects (Fig. 2B–B"). Intriguingly, the overexpression of *E75* not only significantly increased the size of *scrib⁻/⁻,wts⁻/⁻*-induced benign tumors (Fig. 2B,B'), but also astonishingly transformed them into malignant tumors, resulting in 96.3% of the animals being unable to pupate and instead developing into giant larvae (Fig. 2B–B"). This transformation was accompanied by a significant enhancement in tumor invasive ability, as evidenced by the abundant cell autonomous Mmp1 upregulation observed at the leading edge of the invasive tumor cells (Fig. 2C,C'). Additionally, we also observed increased Mmp1 expression and JNK phosphorylation both in tumor clones and invasive tumor cells in a cell-autonomous manner (Fig. 2D,E). Furthermore, *E75* overexpression also synergistically transformed other benign tumors into malignant ones, including *Ras^{V12}* and *Raf^{GOF}, scrib⁻/⁻* (Fig. EV1L–L"). Taken together, these results suggest that ectopic expression of *E75* facilitates the tumor malignant transformation.

## E75 inactivates the Hippo pathway to induce tumor malignancy

To further explore the underlying mechanisms by which *E75* overexpression promotes tumor malignancy, we performed bulk RNA-seq analysis on tumors derived from *scrib⁻/⁻,wts⁻/⁻* and *scrib⁻/⁻,wts⁻/⁻ + E75*. A total of 3457 DEGs were identified, including 1985 upregulated genes and 1472 downregulated genes (Fig. EV2A). We subjected these DEGs to further analysis using the Kyoto Encyclopedia of Genes and Genomes (KEGG) and observed a

significant enrichment and inactivation of the Hippo signaling pathway (Figs. 3A and EV2B,C), a conserved pathway crucial for size control and tumorigenesis (Zheng and Pan, 2019). *Drosophila* core components of the Hippo pathway consist of Hippo (Hpo), Warts (Wts), and Yorkie (Yki). Hpo phosphorylates Wts, which subsequently phosphorylates and inactivates the transcriptional coactivator Yki (Zheng and Pan, 2019). Consistent with its inhibitory role on the Hippo signaling pathway, the ectopic expression of *E75* not only synergistically enhanced *scrib⁻/⁻,wts⁻/⁻*-induced upregulation of Hippo target genes, including *death-associated inhibitor of apoptosis 1 (Diap1)-LacZ* and *expanded (ex)-LacZ* (Fig. 3B), but also led to an increase in clone size (Fig. 3C,C'). Conversely, the depletion of endogenous *E75* resulted in apoptosis (Fig. 3F) and caused a reduction in clone size, as well as adult wing and eye size (Figs. 3C,C' and EV2D), to the same degree as observed during the activation of the Hippo signaling pathway or knockdown of *yki*. Additionally, inhibition of *E75* also suppressed tumor overgrowth induced by *scrib⁻/⁻,wts⁻/⁻* (Figs. 3C,C' and EV2D). Importantly, both overexpression and depletion of *E75* were found to modulate the expression of multiple target genes of the Hippo pathway, such as *Diap1, ex, Cyclin E (CycE)*, and *bantam (ban)* (Figs. 3D and EV2G). Together, these data indicate that E75 negatively regulates Hippo signaling in *Drosophila*.

In line with the observed enrichment of Hippo signaling in E75-induced malignant tumors, the Hippo pathway was also found to be enriched through a bulk RNA-seq analysis performed on E75-overexpressed wing discs (Fig. EV2E,F). To investigate the relationship between E75 and the Hippo pathway components, we conducted a genetic epistasis analysis. We found that overexpression of *E75* induced overgrowth, as well as the upregulation of CycE and Diap1-lacZ were all significantly suppressed upon the knockdown of either *yki* or *scalloped (sd)*, or co-expression of *wts* (Fig. EV2H–I). Similarly, reducing *yki* or *sd* activity significantly blocked the tumor malignancy and pupation defect induced by E75 overexpression in both *Ras^{V12}* and *scrib⁻/⁻,wts⁻/⁻* under pathological conditions (Figs. 3E–E" and EV2J–K'). On the other hand, the depletion of *E75* also impeded the overgrowth phenotype caused by ectopic *yki* expression (Fig. 3F,F'). The mutual genetic dependence of Yki and E75 for their respective functions implies that neither protein functions upstream nor downstream over the other. Hence, these findings collectively suggest that E75 and Yki act genetically interdependent with each other.

Given that E75 possesses a conserved DNA binding domain (DBD) and the ability to initiate gene transcription, we

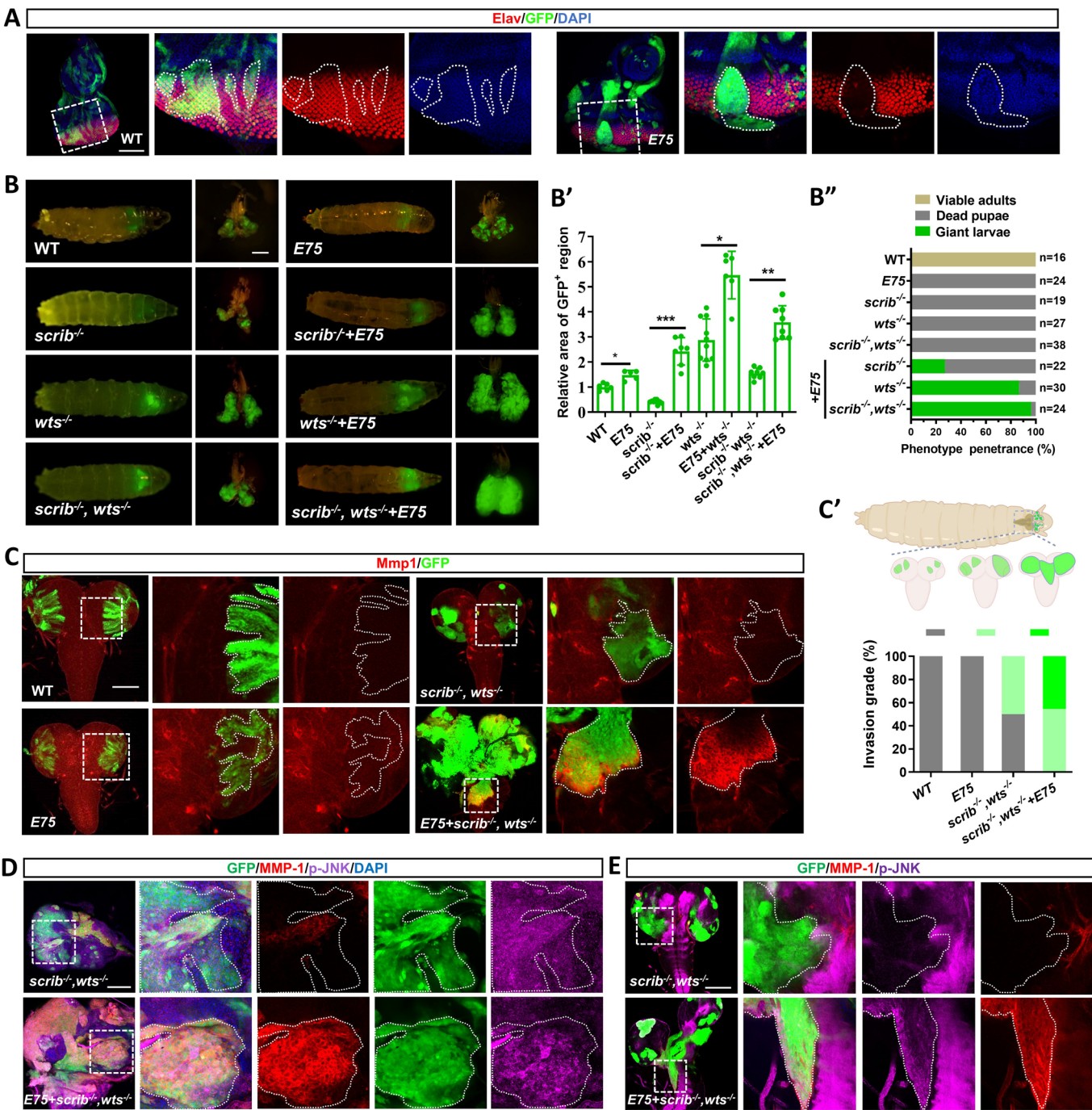

**Figure 2. *E75* overexpression induces tumor malignant transformation.**

(A) Confocal images of eye-antennal discs bearing *ey-Flp*-MARCM-induced mosaics of wild-type and *E75* overexpression stained with anti-β-galactosidase antibody for anti-Elav antibody. Clones are circled by the white dashed line. (B) Dorsal views of *ey-Flp*-MARCM-induced GFP-positive tumor-bearing larvae and the corresponding eye disc or tumor (right). Quantification of relative tumor size (B') (*n* = 6, 5, 9, 7, 10, 6, 8, 8) and larvae pupation rate (B''). Statistical analysis by ordinary one-way ANOVA test; mean ± SD. *n* represents the number of biological replicates. *p* values from left to right, **p* = 0.0358, ****p* = 0.0005, **p* = 0.042, ***p* = 0.0015. (C) Representative confocal images of the ventral nerve cord (VNC) from *ey-Flp*-MARCM-induced tumors with indicated genotype stained with anti-Mmp1 antibody. Quantification of the invasion grade of each genotype (E', bottom panel). Carton illustration of the invasion grade of VNC in tumor-bearing larvae (C', top panel) (*n* = 7, 11, 8, 11). (D, E) Confocal images of eye-antennal discs bearing *ey-Flp*-MARCM-induced tumors (D) and ventral nerve cord (VNC) (E) with indicated genotype stained with anti-Mmp1 and p-JNK antibody. Scale bars: 100 μm (A, C, D, E), 200 μm (B). Source data are available online for this figure.

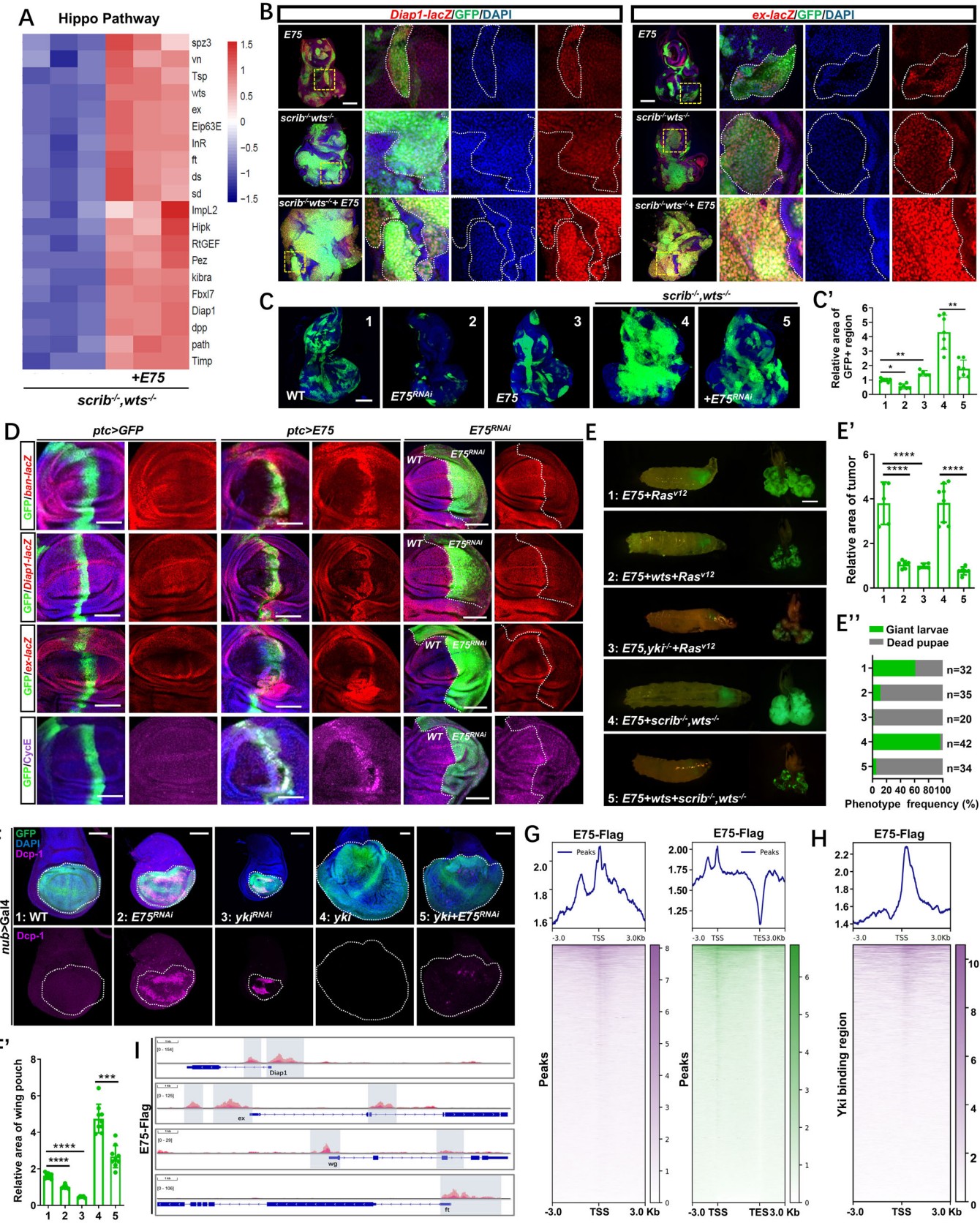

Figure 3.  E75 inactivates the Hippo signaling pathway to induce tumor malignancy.

(A) Heatmap profiles of the expression of Hippo signaling-related genes in *scrib⁻/⁻,wts⁻/⁻* and *E75, scrib⁻/⁻,wts⁻/⁻* tumors. (B) Confocal images of eye-antennal discs bearing *ey-Flp*-MARCM-induced mosaics of indicated genotypes stained with anti-β-galactosidase antibody for *Diap1-LacZ* and *ex-LacZ* staining. (C) Representative images of eye-antennal discs bearing *ey-Flp*-MARCM-induced mosaics of indicated genotypes. (C') Quantification of the relative size of GFP positive regions in (C) (n = 7, 6, 5, 7, 7). n represents the number of biological replicates. Statistical analysis by Brown–Forsythe and Welch ANOVA tests (C'); mean ± SD. p values (C'), from left to right, *p = 0.014, **p = 0.0045, **p = 0.0056. (D) Confocal images of the wing disc with *E75* overexpression under the control of *ptc* promoter and knockdown under the control of *en* or *hh* promoter stained with anti-CycE or anti-β-galactosidase antibodies for *ban-LacZ*, *Diap1-LacZ*, and *ex-LacZ* staining. (E) Dorsal views of *ey-Flp*-MARCM-induced GFP-positive tumor-bearing larvae and the corresponding tumor. Quantification of relative tumor size (E') (n = 5, 6, 4, 8, 6) and larvae pupation rate (E''). n represents the number of biological replicates. Statistical analysis by ordinary one-way ANOVA test (E'); mean ± SD. ****p = 0.0000013, ****p = 0.000005, ****p = 0.00000003. (F) Confocal images of wing disc with indicated genotypes stained with anti-Dcp-1 antibody. (F') Quantification of the relative size of the wing pouch region of F (n = 8, 6, 5, 9, 9). n represents the number of biological replicates. Statistical analysis by Brown–Forsythe and Welch ANOVA tests (F'); mean ± SD. p values (F'), from left to right, ****p = 0.0000045, ****p = 0.00000003, ***p = 0.0001. (G) Binding profiles and heatmaps of E75 CUT&Tag signals are displayed within a region spanning ±3 kb around all canonical transcription start sites (TSS) (left), −3 kb around all TSS, and +3 kb around all canonical transcription end sites (TES) (right). (H) Line plots of the average CUT&Tag signal of E75 peaks (top panel) and heatmaps of the CUT&Tag signals of Yki in *Drosophila* (bottom panel). CUT&Tag signals are displayed within a region spanning ±3 kb around the TSS of all canonical Yki target genes genome-wide. (I) Browser shots of E75 CUT&Tag signal at canonical Hippo pathway target genes. The shaded areas correspond to E75 peaks. Scale bars: 100 µm (B, C, D, F) and 200 µm (E). Source data are available online for this figure.

subsequently performed Cleavage Under Targets and Tagmentation (CUT&Tag) analysis (Kaya-Okur et al, 2019) to investigate the genome-wide DNA binding profiles of E75 in dissected wing pouch regions from *E75* overexpressed discs (Fig. 3G). In addition to the well-established binding motif, we also identified novel binding motifs of *E75* using the HOMER software (Fig. EV2L). By integrating the publicly available chromatin immunoprecipitation sequencing (ChIP-seq) data of Yki-bound chromatin (Nakahashi et al, 2013), we found that the CUT&Tag signals of E75 are significantly enriched in the region where Yki bound (Fig. 3H). Supporting this, our analysis revealed numerous E75 binding sites located within the promoter regions of canonical Hippo target genes, such as *Diap1*, *ex*, *wg*, and *fat* (*ft*) (Fig. 3I). These findings suggest that E75 may serve as a potential partner of Yki in the transcriptional regulation of its downstream genes.

## E75 activates the Notch pathway to promote tumor malignancy

It is important to note that while the activation of Yki or inhibition of Hippo signaling alone can lead to overgrowth, it is insufficient to induce malignancy or the malignant transformation of *scrib⁻/⁻* clones. This suggests that the involvement of additional signaling pathways is essential for E75-induced malignancy. Of note, in addition to *Ras^V12^*, *scrib*-mutated cells could also synergize with the oncogenic form of *Notch* to induce a massive overgrowth (Brumby and Richardson, 2003). Interestingly, upon re-analysis of RNA-seq data obtained from *scrib⁻/⁻,wts⁻/⁻ + E75* tumors, a significant enrichment of the Notch signaling was observed (Fig. EV3A,B). Furthermore, the examination of the binding regions of *Suppressor of Hairless* [*Su(H)*], the transcription factor of the Notch pathway, revealed a strong CUT&Tag signal of E75 (Fig. EV3C). Additionally, several genes known to be targeted by Notch, such as *E(spl)mβ-HLH*, *E(spl)mα-BFM*, *E(spl)m7-HLH*, and *E(spl)m8-HLH*, displayed binding signals of E75 (Fig. 4A). Consistent with these findings, the ectopic expression of *E75* upregulated the expression of several Notch target genes, including *E(spl)mβ-HLH-lacZ*, NRE-GFP, and Cut, in both physiological and E75-induced tumorigenic conditions (Figs. 4B,C and EV3D,E). Conversely, although clonal depletion of *E75* in the eye-antennal disc only partially decreased endogenous Cut expression (Fig. EV3D), inhibition of *E75* in the wing discs notably reduces endogenous Notch signaling activation, as demonstrated by *E(spl)mβ-HLH-lacZ* and Cut staining

(Figs. 4B and EV3F). These results collectively imply that endogenous *E75* plays a more substantial role in regulating Notch activity in the wing discs.

To further investigate the in vivo role of Notch in E75-induced tumor malignancy, we inhibited Notch activation by expressing a dominant negative form of *Notch* (*N^DN^*). Remarkably, although inhibiting *Notch* alone in eye-antennal discs did not significantly affect clone size (Fig. EV3G,G'), inhibiting *Notch* dramatically suppressed the synergistic tumor-promoting effect caused by E75 overexpression and restored the pupation defects (Fig. 4D–D"). Conversely, genetically activating the Notch pathway by co-expressing the activated form of the Notch intracellular domain (*N^act^*) could transform *scrib⁻/⁻,wts⁻/⁻* benign tumors into malignant ones and lead to the development of aggressive tumors (Fig. 4E–F'). Furthermore, we noticed that inhibition of *E75* also impeded N^act^-induced clonal overgrowth (Fig. EV3H,H'). It is important to note that the size of the *N^act^ + wts⁻/⁻* tumor generally appeared larger than that of *N^act^ + scrib⁻/⁻,wts⁻/⁻* tumor, possibly due to the two-dimensional expansion of tumor cells without disrupting the single-layer polarity architecture of the eye epithelium. These findings collectively suggest that the activation of the Notch pathway is both necessary and sufficient for the malignant transformation of E75-induced tumor progression.

## *Myc* is required for E75-induced malignant tumor transformation

The *Myc* oncogene plays a critical role in the progression of various human cancers. Intriguingly, *Myc* also serves as an evolutionarily conserved transcriptional target for both the Hippo and Notch signaling pathways (Neto-Silva et al, 2010; Weng et al, 2006). Given this, we further explored whether Myc is a possible downstream target in tumor transformation induced by E75 overexpression. Depletion of *E75* decreased endogenous *Myc* transcription (Fig. EV3I), whereas ectopic expression of *E75* caused a robust increase in both transcription and protein levels of *Myc*, in a Yki- and Notch-dependent manner (Fig. 4G,H). In line with this, our CUT&Tag analysis identified multiple E75 binding sites on the promoter and gene body region of *Myc* (Fig. EV3J). Moreover, the depletion of *Myc* not only suppressed the tissue overgrowth phenotype caused by *E75* overexpression in the developing wing (Fig. EV3K–L'), but also significantly blocked E75-induced malignant transformation of

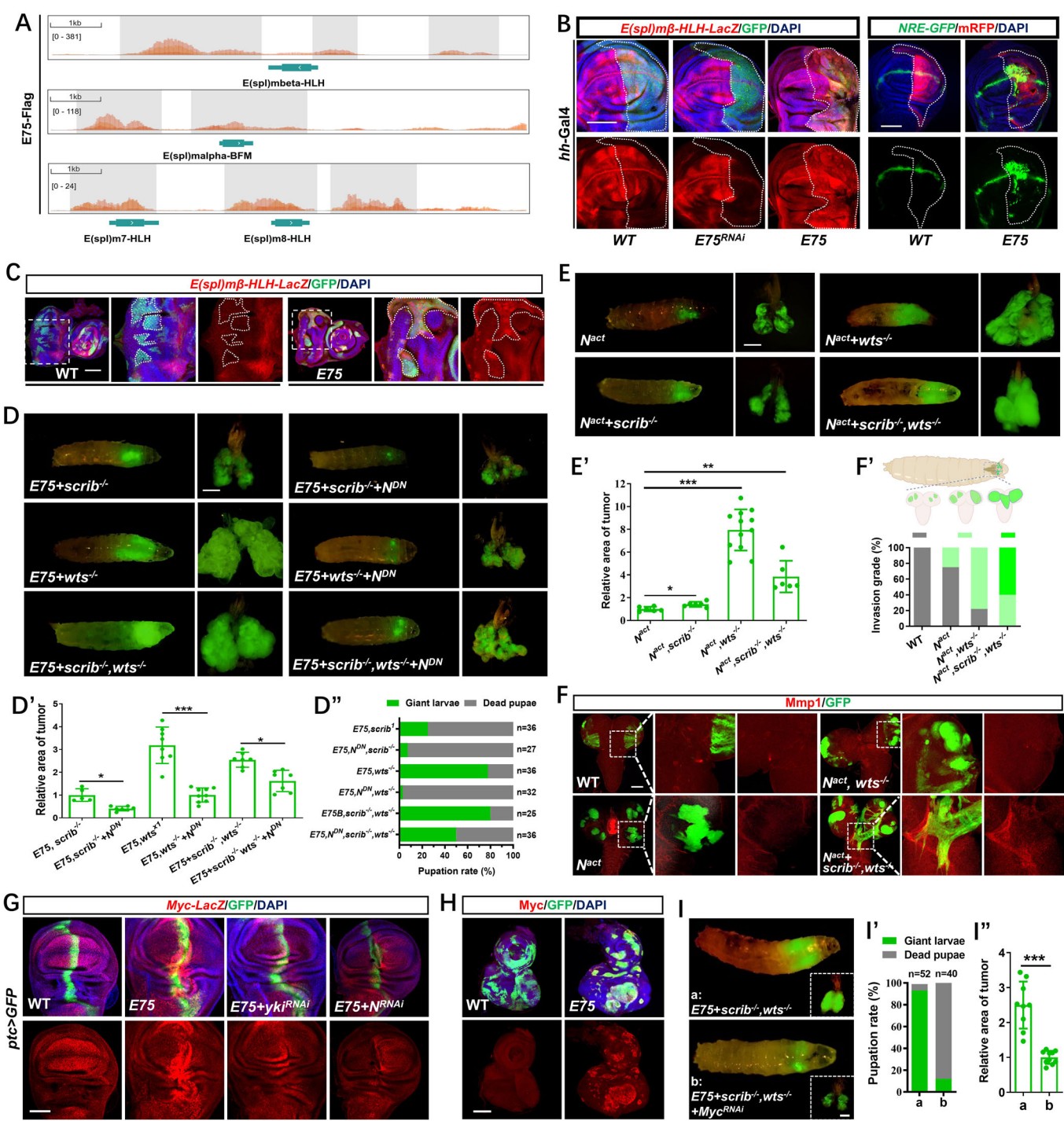

scrib$^{-/-}$,wts$^{-/-}$ tumors (Fig. 4I–I"). These data collectively demonstrate that Myc functions as a downstream target gene of E75, influencing both tumor growth and malignancy.

## E75 integrates the Hippo and Notch pathways at the transcriptional factor level

The observed similarity in binding sites between E75 and Yki or Su(H), as demonstrated by our E75 CUT&Tag analysis, indicates a

potential physical interaction between E75 and the transcriptional complexes associated with the Hippo and Notch pathways. To test this, we performed a proximity ligation assay (PLA) in the Drosophila wing and eye epithelium to detect the protein-protein interactions at the subcellular level in situ (Fredriksson et al, 2002). Compared to the negative controls, we detected robust positive PLA signals between Flag-E75 and both Myc-tagged Yki and HA-tagged Sd (Fig. 5A,B). The interactions between E75 and Sd were also seen in eye-disc clones expressing E75$^{Flag}$, yki, and Sd$^{HA}$

**Figure 4. E75 activates the Notch pathway to promote tumor malignancy.**

(A) Browser shots of E75 CUT&Tag signal at canonical Notch pathway target genes. The shaded areas correspond to E75 peaks. (B) Confocal images of wing disc with indicated genotypes stained with anti-β-galactosidase antibody for *E(spl)mβ-HLH-LacZ* staining (left) or *NRE-GFP* expression (right). (C) Eye-antennal discs bearing *ey-Flp*-MARCM-induced mosaics of wild-type and *E75* overexpression stained with anti-β-galactosidase antibody for the *E(spl)mβ-HLH-LacZ* staining. (D, E) Dorsal views of *ey-Flp*-MARCM-induced GFP-positive tumor-bearing larvae and the corresponding eye disc or tumor (right). Quantification of relative tumor size (D′, n = 5, 7, 8, 9, 7, 7; E′, n = 6, 6, 12, 6), and larvae pupation rate (D″). n represents the number of biological replicates. Statistical analysis by Brown–Forsythe and Welch ANOVA tests (D′, E′); mean ± SD. p values (D′ from left to right), *p = 0.0414, ***p = 0.0005, *p = 0.0123. p values (E′ from left to right), *p = 0.041, ***p = 0.0007, **p = 0.0014. (F) Representative confocal images of VNC from *ey-Flp*-MARCM-induced tumors with indicated genotypes stained with anti-Mmp1 antibody. Carton illustration of invasion grade of VNC in tumor-bearing larvae (F′, top panel). Quantification of invasion percentage of each genotype (F′, bottom panel) (n = 12, 8, 10, 9). (G) Representative confocal images of wing disc bearing indicated genotypes stained with anti-β-galactosidase antibody for *Myc-LacZ* staining. (H) Eye-antennal discs bearing *ey-Flp*-MARCM-induced mosaics of wild-type and *E75* overexpression stained with anti-Myc antibody. (I) Dorsal views of *ey-Flp*-MARCM-induced GFP-positive tumor-bearing larvae and the corresponding eye disc or tumor (right corner). Quantification of larvae pupation rate (I′) and relative tumor size (I″) (n = 9, 10). n represents the number of biological replicates. Statistical analysis by Student's t-test (I″); mean ± SD. ***p = 0.0003. Scale bars: 100 μm (B, C, F, G, H) and 200 μm (D, E, I). Source data are available online for this figure.

(Fig. EV4A). Additionally, PLA revealed strong interactions between overexpressed E75 and the endogenous NICD and Su(H) (Fig. 5C). To confirm these physical interactions, we performed co-immunoprecipitation (co-IP) assays with E75 and the mentioned transcription factors. Consistently, when *E75* was ectopically expressed with *Yki* and *Sd* in the developing eye epithelium, robust physical interactions were detected between Flag-tagged E75 and Yki or HA-tagged Sd (Fig. 5D,E). Similarly, both endogenous NICD and Su(H) could form a complex with Flag-tagged E75 in tumor clones that co-expressed *E75*, *NICD*, and *Yki* (Fig. 5F,G). Together, these data suggest that E75 can physically bind with the downstream transcriptional complexes of both Hippo and Notch pathways.

Under tumorigenic stress, NICD can form a complex with a transcriptional coactivator with PDZ-binding motif (TAZ), the human ortholog of Yki (Kim et al, 2017). Indeed, we noticed that discs expressing both *yki* and *N^{act}* showed a detectable but moderate PLA signal (Fig. EV4B). Interestingly, it is noteworthy that no PLA signals were detected between Su(H) and Yki or Sd under physiological conditions (Fig. EV4C). However, upon the over-expression of *E75*, a dramatic increase in PLA signals was observed between Su(H) and Yki or Sd, as well as NICD and Sd (Figs. 5H and EV4B). This observation suggests that ectopic expression of *E75* can enhance the physical association between downstream TFs of the Hippo and Notch pathways. Consistent with this notion, ectopic expression of *E75* synergistically enhanced the tumorigenic potential of the benign tumor induced by the co-expression of *N^{act}* and *yki*, leading to their transformation into malignant tumors (Figs. 5I and EV4D).

To further investigate the effect of E75 on the TF interacting network and chromatin landscape, we conducted NICD and Yki CUT&Tag analyses on tumor samples from *N^{act} + yki* expressing (Weghorn and Sunyaev, 2017) and *E75 + N^{act} + yki* expressing (ENY) eye epithelium. The peaks were annotated based on the genomic locations (Fig. EV4E) and relative distance to transcriptional start sites (TSSs) (Fig. EV4F). For the NY tumors, a total of 6825 NICD-annotated and 7895 Yki-annotated peaks were identified, while for the ENY tumors, 5758 NICD-annotated and 6756 Yki-annotated peaks were uncovered. To visualize the levels and distribution of NICD and Yki, we generated spike-normalized coverage heatmaps at a genome-wide scale (Fig. EV4G). Interestingly, we observed a general decrease in signal intensities of both NICD and Yki around the TSSs when *E75* was co-expressed

(Fig. EV4H). Next, we plotted the NICD and Yki CUT&Tag signals around the TSS of protein-coding genes where E75 bound. We found that while both NICD and Yki bind to the regions accessible to E75, their signal intensities were also reduced upon *E75* expression (Fig. 5J).

Taken together, these results suggest that E75 effectively integrates the Hippo and Notch pathways at the transcriptional level. Ectopically expressed E75 not only physically associates with the TFs of both pathways, but also facilitates its binding to novel chromatin regions. Both actions collectively contribute to tumor malignancy.

## Silencing *NR1D2* suppressed glioblastoma stem cell-driven tumor growth

Next, we investigated whether E75 plays a conserved role in regulating Notch and Hippo pathway-mediated tumorigenesis in mammals. *NR1D2* (*nuclear receptor subfamily 1 group D member 2*) encodes the mammalian ortholog of *E75* and belongs to the nuclear hormone receptor family. We have previously demonstrated that depleting *NR1D2* impedes the cell mobility and viability of glioblastoma (GBM) - a highly aggressive and fatal brain tumor (Yu et al, 2018). Interestingly, out of the 3760 DEGs identified in *NR1D2*-depleted GBM cells, both the Notch and Hippo signaling pathways were enriched through KEGG and GO analyses (Fig. EV5A). Furthermore, the functional annotation analysis of the DEGs revealed an enrichment of 15 hormone-related terms (Fig. EV5B). This set of 201 DEGs consisted of 69 upregulated genes and 132 downregulated genes. Notably, the knock-down of *NR1D2* significantly affected two major hormone categories: steroid hormones and thyroid hormones (Fig. EV5B). Moreover, analysis utilizing GEPIA (Gene Expression Profiling Interactive Analysis) indicated that several target genes of both pathways positively correlated with elevated *NR1D2* expression in GBM (Fig. EV5C).

GBM exhibits intra-tumoral heterogeneity with self-renewing glioblastoma stem cells (GSCs) at the hierarchical apex, and GSCs are functionally characterized by their capacities for self-renewal and tumor initiation, along with their additional roles in tumor angiogenesis, radio-resistance, and chemoresistance (Bao et al, 2006; Gimple et al, 2022). Notably, we found the expression level of *NR1D2* was significantly upregulated in GSCs compared to neural stem cells (NSCs) (Fig. 6A) (Mack et al, 2019). Furthermore, the depletion of *NR1D2* using two non-overlapping short hairpin

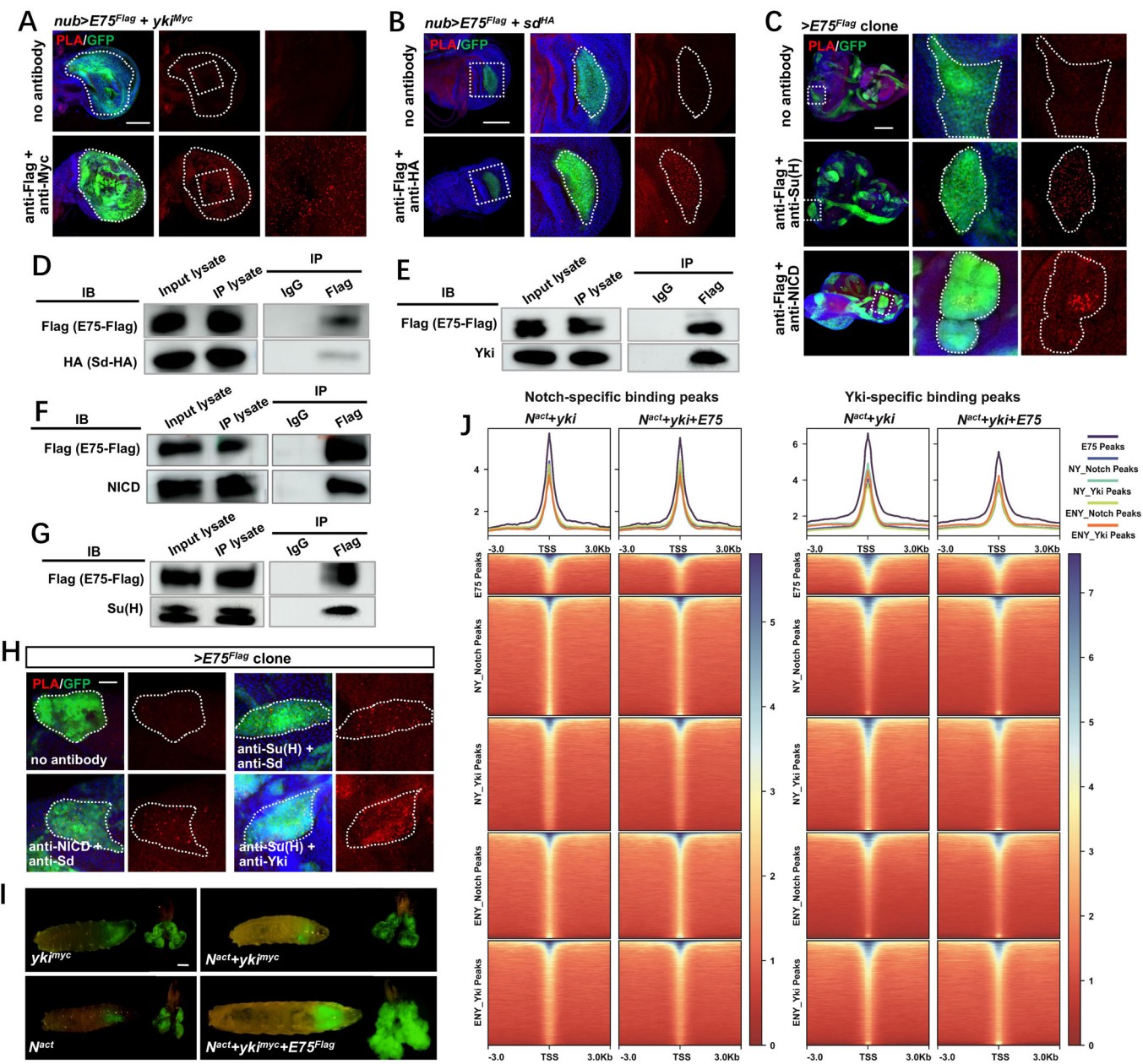

**Figure 5. E75 integrates the Hippo and Notch pathways at the transcription factor level.**

(A, B) Proximity ligation assay (PLA) was performed on wing discs with indicated genotypes to test close-proximity interactions between Flag-tagged E75 and Myc-tagged Yki (A) or Flag-tagged E75 and HA-tagged Sd (B). (C) PLA was performed on *ey-Flp*-MARCM-induced GFP-positive clones with indicated genotypes to evaluate close-proximity interactions between E75 and Su(H) or NICD. (D–G) Co-IP assays to detect in vivo physical interactions between *E75^Flag* and *Sd^HA*, NICD, Yki, and Su(H). Lysates from *ey-Flp*-MARCM-induced epithelial tumors with indicated genotypes (D and E: ey-Flp1/+; act>y+>GAL4, UAS-GFP/UAS-Sd^HA, UAS-E75^Flag; Tub-Gal80, FRT82B/FRT82B, UAS-Yki; F and G: ey-Flp1/+; act>y+>GAL4, UAS-GFP/UAS-Notch^act, UAS-E75^Flag; Tub-Gal80, FRT82B/FRT82B, UAS-Yki) were immunoprecipitated and probed with the indicated antibodies. (H) PLA was performed on *ey-Flp*-MARCM-induced *E75^Flag* overexpression clones to test close-proximity interactions between endogenous NICD and Sd, Su(H) and Sd, Su(H), and Yki. (I) Representative dorsal views of ey-Flp-MARCM-induced GFP-positive tumor-bearing larvae and the corresponding dissected tumor (right). (J) Line plots (top panels) show the average CUT&Tag signal of Notch-specific peaks of $N^{act} + Yki$ and $E75 + N^{act} + Yki$, and Yki-specific peaks of $N^{act} + Yki$ and $E75 + N^{act}+Yki$. Clustered heatmaps (bottom panels) show the Notch-specific peaks of $N^{act} + Yki$ and $E75 + N^{act} + Yki$ (left) and Yki-specific peaks of $N^{act} + Yki$ and $E75 + N^{act} + Yki$ (right). CUT&Tag signals are displayed within a region spanning ±3 kb around all canonical TSS. E for E75, N for Notch, and Y for Yki. Scale bars: 20 um (G), 100 μm (A–C), and 200 um (I). Source data are available online for this figure.

RNAs (shRNA) (Fig. EV5D) strongly decreased cell proliferation of two patient-derived GSCs (MGG6 and MGG4) (Figs. 6B and EV5E) and led to a significant decrease in the expression levels of Notch and Hippo pathway target genes in GSCs (Fig. 6C,D). Similar to the interacting network of TFs identified in *Drosophila*, we found that NR1D2 also physically interacts with TFs of both the Hippo and Notch pathways in GSCs and GBM cells, as demonstrated by PLA and co-IP assays (Figs. 6E–G and EV5F,F'). To verify whether

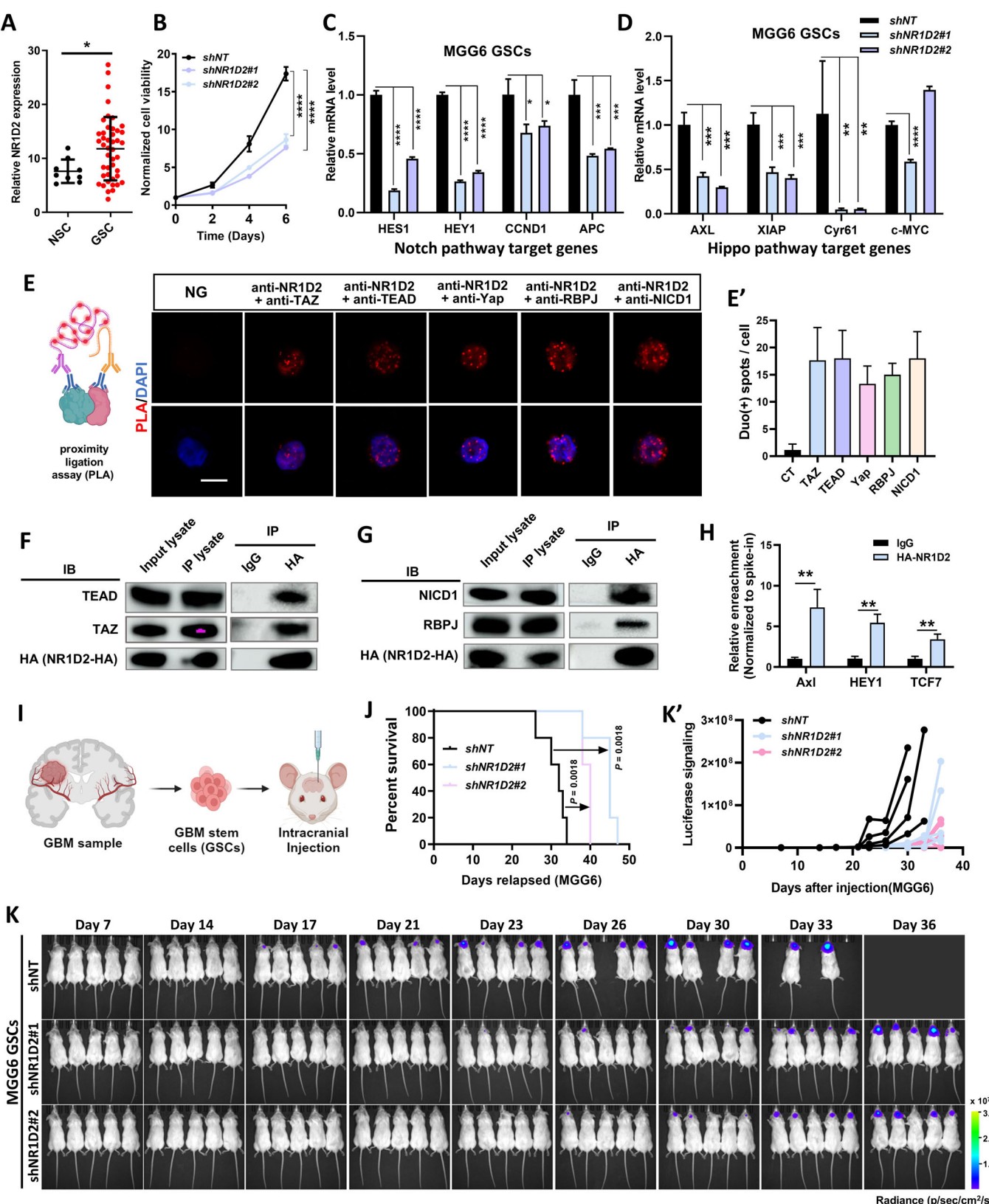

**Figure 6. Silencing *NR1D2* suppressed glioblastoma stem cell-driven tumor growth.**

(A) Relative expression of *NR1D2* in neural stem cell (NSC) and glioblastoma stem cell (GSC). Statistical analysis by unpaired Student's *t*-tests (A); mean ± SD. *$p = 0.0418$. (B) Relative cell viability of MGG6 of shNT and *shNR1D2*. Statistical analysis by Two-way repeated measures ANOVA with Dunnett's multiple hypothesis test correction; mean ± SD. ****$p = 0.0000068$, ****$p = 0.00000206$. (C, D) Relative mRNA level of Notch target genes (*HES1, HEY1, CCND1,* and *APC*) (C) and Hippo target genes (*Axl, XIAP, Cyr61,* and *Myc*) (D). Statistical analysis by Ordinary one-way ANOVA tests (C, D); mean + SD. *p* values in (C) from left to right, ****$p = 0.00000002$, ****$p = 0.00000047$, ****$p = 0.00000001$, ****$p = 0.000000003$, *$p = 0.0101$, *$p = 0.0249$, ***$p = 0.0004$, ***$p = 0.0008$. *p* values in (D) from left to right, ***$p = 0.0003$, ***$p = 0.0001$, ***$p = 0.0007$, ***$p = 0.0004$, **$p = 0.0061$, **$p = 0.006$, ****$p = 0.00000048$. (E) PLA was performed in MGG6 GSCs to test close-proximity interactions between NR1D2 and TAZ, TEAD, Yap, RBPJ, and NICD1. (E') Quantification of PLA signal intensity in (E) ($n = 7, 6, 6, 6, 7$); mean + SD. (F, G) Co-IP assays to detect physical interactions between NR1D2-HA and endogenous TEAD, TAZ, NICD1, and RBPJ in U87 MG cells. Lysates from U87 MG cells with stably transfected NR1D2-HA were immunoprecipitated (IP) and probed with the indicated antibodies. (H) CUT&Tag qPCR analysis of *Axl, HEY1,* and *TCF7* in MGG6. MGG6 cells transfected with *HA-NR1D2* were used for HA enrichment quantification on promoter region ($-500$ to $0$). $n = 3$ for each sample. *n* represents the number of technical replicates. Statistical analysis by two-tailed Student's *t*-tests; mean + SD. *p* values, from left to right, **$p = 0.0078$, **$p = 0.0027$, **$p = 0.0043$. (I) Schematic diagram of GSCs brain injection. (J) Survival curve of NSG mice bearing intracranial tumors from MGG6 GSCs transfected with *shNT, shNR1D2#1,* or *shNR1D2#2*, respectively. Statistical analysis by Log-rank (Mantel-Cox) test, **$p = 0.0018$ (*shNT* vs. *shNR1D2#1*), **$p = 0.0018$ (*shNT* vs. *shNR1D2#2*). (K) In vivo bioluminescence imaging of NSG mice bearing tumors on the 7th, 14th, 17th, 21st, 23rd, 26th, 30th, 33rd, and 36th day after MGG6 GSCs were injected into the mice brains. The injected MGG6 GSCs were transfected with *shNT, shNR1D2#1,* or *shNR1D2#2* respectively. (K') Quantification of tumor size by in vivo luciferase assays. Scale bar: 10 μm (E). Source data are available online for this figure.

NR1D2 could directly bind to the promoter regions of Hippo and Notch target genes, we performed chromatin immunoprecipitation (ChIP) assays in GSCs and GBM cells that stably expressed HA-tagged NR1D2 (HA-NR1D2). Our data showed that NR1D2 binds to the promoter regions of multiple Hippo and Notch target genes, including *Axl, HEY1,* and *TCF7* (Figs. 6H and EV5G). Finally, we examined the impact of disrupting *NR1D2* on GSC-driven tumor growth in vivo. Luciferase-expressing GSCs, transfected with lentivirus expressing *NR1D2* shRNA or non-targeting control shRNA (shNT), were injected into the right cerebral cortex of NSG mouse brains (Fig. 6I). Bioluminescent imaging showed that *NR1D2* knockdown markedly suppressed GSC-driven tumor growth and extended the survival of mice compared with controls (Figs. 6J–K' and EV5H–I). Collectively, these results indicate that NR1D2 plays a conserved role in regulating the Hippo and Notch pathways and is crucial for GSC-induced tumor growth in vivo.

## Discussion

Hormone therapies are commonly employed to inhibit the proliferation of hormone-dependent cancers, such as breast and prostate cancers. Nonetheless, as time passes, cancer cells may develop mechanisms to survive, even in the absence of hormones or in the face of hormone blockers, resulting in resistance to hormone therapy (Hanker et al, 2020; Metcalfe et al, 2018). However, the specific mechanisms responsible for tumor progression and malignancy in hormone signaling-reduced tumors remain predominantly uninvestigated. Here we identified the primary hormone-responsive gene *E75* as a crucial oncogene in promoting tumor malignant transformation in *Drosophila*. We found that the inhibition of ecdysone signaling is a key characteristic of *Drosophila* malignant tumors. By gain-of-function analysis of ecdysone-induced genes, we found ectopic expression of *E75* could feedback inhibit ecdysone signaling and facilitate the tumor malignancy of benign tumors by integrating two essential tumor-regulating pathways, namely the Hippo and Notch pathways (Fig. 7).

The nuclear receptor E75 has been implicated in the regulation of various biological processes, including axon degeneration and regrowth (Rabinovich et al, 2016), circadian clock regulation

(Jaumouillé et al, 2015), and cell migration (Wang et al, 2020). Recent studies have shown that ecdysone, released from the adult ovary following mating, exerts a remote effect on intestinal cell differentiation and proliferation through the downstream response gene *E75* (Ahmed et al, 2020; Zipper et al, 2020). In this study, we reveal a novel growth and tumor-regulating role of *E75* in *Drosophila* epithelium cells. Under physiological conditions, depletion of *E75* induces apoptosis and leads to size reduction, while *E75* overexpression promotes overgrowth. This is in line with a recent study showing that low titer ecdysone signaling activation promotes cell proliferation (Gavish et al, 2023). Under pathological conditions, we found that *E75* overexpression induces tumor malignant transformation in diverse tumor models. Moreover, consistently with the growth suppression role of ecdysone activation, we also show that hyperactivation of different EcR isoforms effectively impedes $lgl^{-/-}, Ras^{V12}$-induced tumor malignancy.

The physical associations observed between E75, and the downstream transcription complexes of the Hippo and Notch pathways indicate that E75 potentially functions as an integrator, bridging these two signaling networks. We believe that this E75-mediated integration enables cells to generate coordinated responses that are essential for various physiological and pathological processes. Through the integration of CUT&Tag with RNA-seq analyses, we found that both NICD and Yki bind to the regions accessible to E75. Notably, E75 also specifically binds and induces the expression of numerous additional genes, indicating its potential role as an integrator in coordinating the activity of multiple signaling pathways. Further investigations are required to gain a comprehensive understanding of the underlying mechanisms involved.

Our data indicates that both *E75* and *NR1D2* should be considered oncogenes conceptually. The expression of *E75* promotes tumor growth, and so does *NR1D2*. *E75*, being an ecdysone-induced gene, is downregulated in *Drosophila* malignant tumors due to the reduction of ecdysone signaling. Currently, our understanding of why ecdysone signaling is downregulated in fly malignant tumors is limited. However, it would be an oversimplification to conclude that the expression of every oncogene must be upregulated in tumors. Similarly, multiple target genes of Yki/YAP act as upstream regulators of the Hippo pathway and

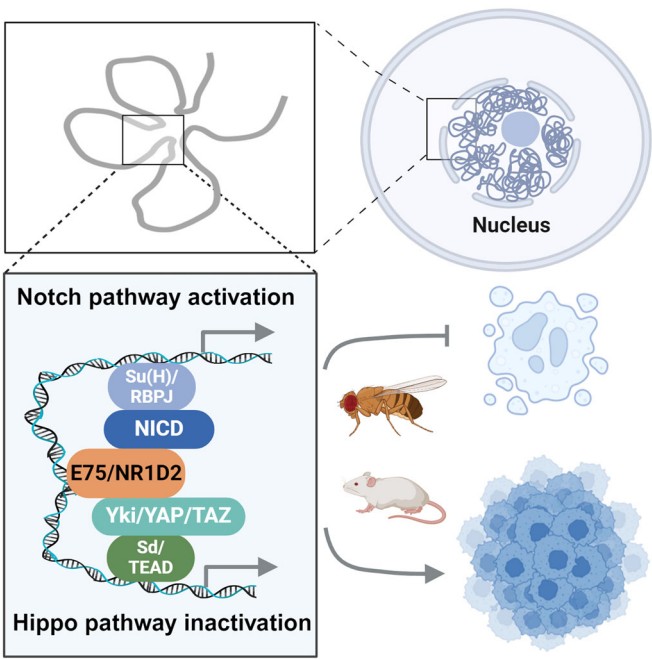

**Figure 7. Schematic of E75/NR1D2 induced tumor malignancy in *Drosophila* and mammals.**

This model illustrates the molecular mechanisms by which E75 and NR1D2 contribute to tumorigenesis. E75 overexpression supports tumor growth, and its suppression triggers cell death. E75 modulates both Hippo and Notch pathways by binding with their transcriptional effectors—Sd, Yki, NICD, and Su(H) in *Drosophila*; TEAD, YAP/TAZ, RBPJ, and NICD in mammals—at a transcriptional and chromatin level. This interaction activates oncogenic genes within these pathways, promoting the malignant transformation of tumors.

should be classified as tumor suppressors. Nevertheless, their levels are still upregulated when Yki/YAP is activated, which is a common phenomenon observed in various tumors.

Previous studies have investigated the function of NR1D2, also known as REV-ERBβ, the mammalian homolog of E75, in GBM and melanocytic navy. However, these studies have reported inconsistencies of NR1D2 in the regulation of tumor progression (Sulli et al, 2018; Yu et al, 2018). Additionally, it is noteworthy that many studies employed NR1D2 agonists to examine its function, but this approach may have induced NR1D2-independent effects on cell proliferation (Dierickx et al, 2019; Xu et al, 2022). Consequently, the validation of *NR1D2* in vivo loss-of-function becomes essential for a better understanding of its potential as a therapeutic target and its association with tumor progression. Consistent with the elevation of *NR1D2* expression in the GSCs, we found that the depletion of *NR1D2* inhibits GSC proliferation and improves the survival of tumor-bearing mice. This indicates that *NR1D2* functions as an oncogene in promoting GBM progression in vivo. Moreover, we show that NR1D2, similar to E75, physically interacts with transcription factors of both Hippo and Notch pathways, regulating the expression of their target genes. This suggests that E75/NR1D2 has an evolutionarily conserved function in tumorigenesis. While hormone therapy is not conventionally considered as a standard treatment for GBM, recent studies have presented its potential application in targeting nuclear hormone receptors for GBM treatment (Gonzalez-Mora and Garcia-Lopez,

2021; Rodríguez-Lozano et al, 2019). Further investigation of the in vivo functions of *NR1D2* using knock-out mice across various tumor types could be particularly insightful and advance the discovery of novel therapeutic strategies for both hormone-dependent and hormone-independent tumors.

## Methods

### Reagents and tools table

| Reagent or Resource | Source | Identifier |
|---|---|---|
| **Experimental models: Drosophila melanogaster** | | |
| w[1118] | Bloomington Drosophila Stock Center | Cat# 5905 |
| UAS-E75.RNAi (II) | Vienna Drosophila Resource Center | Cat# v44851 |
| UAS-E75.RNAi (III) | Bloomington Drosophila Stock Center | Cat# 26717 |
| ptc-Gal4 | Gift from Lei Xue (Tongji University) | FBal0287777 |
| hh-Gal4 | Gift from Lei Xue (Tongji University) | FBal0121962 |
| en-Gal4 | Bloomington Drosophila Stock Center | Cat# 99568 |
| nub-GAL4 | Bloomington Drosophila Stock Center | Cat# 86108 |
| UAS-E75[Flag] | Gift from Jiong Chen (Nanjing University) | Original source: (Rabinovich et al, 2016) |
| mir-ban[L1170a] (ban-LacZ) | Bloomington Drosophila Stock Center | Cat# 10154 |
| ex[e1] (ex-lacZ) | Bloomington Drosophila Stock Center | Cat# 44249 |
| myc-LacZ[G0354]/ FM7c | Bloomington Drosophila Stock Center | Cat# 11981 |
| E(spl)-HLH-mβ-lacZ | Gift from Hai Huang (Zhejiang University) | N/A |
| NRE-GFP | Gift from Hai Huang (Zhejiang University) | N/A |
| yki[B5] | Gift from Duojia Pan (University of Texas Southwestern Medical Center) | (Huang et al, 2005) FBal0194179 |
| UAS-yki[Myc] (II) | Gift from Lei Xue (Tongji University) | N/A |
| UAS-yki (III) | Gift from Duojia Pan (University of Texas Southwestern Medical Center) | (Huang et al, 2005) |
| UAS-sd.RNAi | Gift from Lei Zhang (Chinese Academy of Sciences) | (Zhang et al, 2008a) |
| UAS-Hr3 | Fly ORF | Cat# F000034 |
| UAS-ftz-f1 | Bloomington Drosophila Stock Center | Cat# 64290 |
| UAS-Eip93F | Fly ORF (Gift from Sheng Li lab) | Cat# F000587 |
| UAS-Br-Z1 | Bloomington Drosophila Stock Center | Cat# 51190 |
| UAS-Sd[HA] | Gift from Lei Zhang (Chinese Academy of Sciences) | N/A |
| UAS-wts | Gift from Shian Wu (Nankai University) | N/A |
| UAS-Notch.RNAi | Vienna Drosophila Resource Center | Cat# v1112 |
| UAS-myc.RNAi | Vienna Drosophila Resource Center | Cat# v2947 |
| UAS-Notch[act] | Gift from Hai Huang (Zhejiang University) | N/A |
| UAS-Notch[DN] | Gift from Hai Huang (Zhejiang University) | N/A |
| UAS-Yki.RNAi | Vienna Drosophila Resource Center | Cat# v40497 |
| EcRE-lacZ | Bloomington Drosophila Stock Center | Cat# 4516 |
| EcRE-lacZ | Bloomington Drosophila Stock Center | Cat# 4517 |
| UAS-Ras[V12], FRT82B, scrib[1]/ TM6B Tb | Gift from Tian Xu (Westlake University) | N/A |

| Reagent or Resource | Source | Identifier |
|---|---|---|
| UAS-Ras$^{V12}$ | Previously described | (Ma et al, 2017) |
| UAS-GFP | Gift from Lei Xue (Tongji University) | N/A |
| GMR-GAL4 (II) | Gift from Lei Xue (Tongji University) | N/A |
| UAS-EcRA | Bloomington Drosophila Stock Center | Cat# 6470 |
| UAS-EcRB1 | Bloomington Drosophila Stock Center | Cat# 6469 |
| UAS-EcRC | Bloomington Drosophila Stock Center | Cat# 6868 |
| lgl$^4$, FRT40A, UAS-Ras$^{V12}$/Cyo | Gift from Tian Xu (Westlake University) | N/A |
| UAS-Raf$^{GOF}$, FRT82B, srcib$^1$/TM6B Tb | Gift from Tian Xu (Westlake University) | N/A |
| FRT40A | Gift from Tian Xu (Westlake University) | N/A |
| FRT82B | Gift from Tian Xu (Westlake University) | N/A |
| scrib$^1$ | Gift from Tian Xu (Westlake University) | Original source: (Bilder et al, 2000) |
| wts$^{x1}$ | Gift from Tian Xu (Westlake University) | (Xu et al, 1995) |
| yw, ey-Flp1; act>y$^+$>GAL4, UAS-GFP; FRT82B, Tub-Gal80, (82B Tester) | Gift from Tian Xu (Westlake University) | N/A |
| yw, ey-Flp;Tub-Gal80, FRT40A; act>y$^+$>Gal4, UAS-GFP | Gift from Tian Xu (Westlake University) | N/A |
| **Experimental models: Mouse** | | |
| NOD.Cg-Prkdc$^{scid}$Il2rg$^{em1Smoc}$ | Shanghai Model Organisms Center, Inc. | Cat# NM-NSG-001 |
| **Experimental models: Antibodies** | | |
| Mouse monoclonal anti-Broad | Developmental Studies Hybridoma Bank | Cat# 25E9 AB_3086638 |
| Mouse monoclonal anti-β-gal | Developmental Studies Hybridoma Bank | Cat# 40-1a; RRID: AB_2314509 |
| Rat anti-Elav | Developmental Studies Hybridoma Bank | Cat# 7E8A10 AB_3086637 |
| Mouse monoclonal anti-Mmp1 | Developmental Studies Hybridoma Bank | Cat# 3A6B4; RRID: AB_579780, Cat# 3B8D12; RRID: AB_579781, Cat# 5H7B11; RRID: AB_579779 |
| Rabbit polyclonal anti-Cleaved Drosophila Dcp-1 | Cell Signaling Technology | Cat# 9578; RRID: AB_2721060 |
| Rabbit polyclonal anti-active JNK | Promega | Cat# v2973; RRID: AB_3391716 |
| Rabbit polyclonal anti-cyclin E | Santa Cruz | Cat#sc-33748 AB_638863 |
| Mouse monoclonal anti-NICD | Developmental Studies Hybridoma Bank | Cat# C17.9C6 RRID: AB_528410 |
| Mouse anti-Cut homeobox (Cut) | Developmental Studies Hybridoma Bank | Cat# 2B10 RRID: AB_528186 |
| Mouse monoclonal anti-Wg | Developmental Studies Hybridoma Bank | Cat# 4D4 RRID: AB_528512 |
| Rabbit anti-Sd | Gift from Lei Zhang (Chinese Academy of Sciences) | (Jin et al, 2013) |
| Rabbit monoclonal anti-DYKDDDDK Tag | Cell Signaling Technology | Cat# 14793; RRID: AB_2572291 |

| Reagent or Resource | Source | Identifier |
|---|---|---|
| Mouse monoclonal anti-Flag | Sigma-Aldrich | Cat# F1804 RRID: AB_262044 |
| Rabbit monoclonal anti-HA-Tag | Cell Signaling Technology | Cat# 3724; RRID: AB_1549585 |
| Mouse monoclonal anti-HA-Tag | Proteintech | Cat#: 66006-2-Ig RRID: AB_ 2881490 |
| Mouse monoclonal anti-Myc-tag | Cell Signaling Technology | Cat# 2276; RRID: AB_331783 |
| Mouse monoclonal anti-NR1D2 | Santa Cruz | Cat# sc-100911 RRID: AB_2154757 |
| Mouse anti-dMyc | Developmental Studies Hybridoma Bank | Cat# P4C4-B10 RRID: AB_2753231 |
| Mouse monoclonal anti-Su(H) | Santa Cruz | Cat# sc-398453 RRID: AB_3086636 |
| Rabbit polyclonal anti-Yki | Gift from Duojia Pan (UT Southwestern Medical Center) | (Dong et al, 2007) |
| Rabbit polyclonal anti-RBPSUH (RBPJ) | Cell Signaling Technology | Cat# 5442, RRID: AB_10695407 |
| Rabbit monoclonal anti-cleaved Notch1 | Cell Signaling Technology | Cat# 4147 RRID: AB_2153348 |
| Rabbit monoclonal anti-Pan-TEAD | Cell Signaling Technology | Cat#13295 RRID: AB_2687902 |
| Rabbit monoclonal anti-Yap | Cell Signaling Technology | Cat# 14074 RRID: AB_2650491 |
| Rabbit monoclonal anti-TAZ | Cell Signaling Technology | Cat#70148 RRID: AB_3086635 |
| Rabbit IgG | Beyotime | Cat#A7016 RRID: AB_2905533 |
| Goat anti-Mouse IgG (H + L) Alexa Fluor™ Plus 555 | Invitrogen | Cat# A32727; RRID: AB_2633276 |
| Goat anti- IgG (H + L) Alexa Fluor™ Plus 555 | Invitrogen | Cat# A32732; RRID: AB_2633281 |
| Goat anti- IgG (H + L) Alexa Fluor™ Plus 647 | Invitrogen | Cat# A32733; RRID: AB_2633282 |
| Goat anti-Mouse IgG H&L | Abcam | Cat #ab6702 RRID: AB_956012 |
| Goat anti-Rabbit IgG H&L | Abcam | Cat #ab6708 RRID: AB_956005 |
| Anti-rabbit IgG, HRP-linked Antibody | Cell Signaling Technology | Cat 7074 RRID: AB_2099233 |
| Anti-mouse IgG, HRP-linked Antibody | Cell Signaling Technology | Cat #7076 RRID: AB_330924 |
| **Critical commercial assays** | | |
| Protease inhibitor cocktail | Sigma | Cat# P8340 |
| Hyperactive Universal CUT&Tag Assay Kit | Vazyme | Cat# TD904 |
| Duolink™ In Situ | Sigma-Aldrich | Cat# DUO92008-100RXN; DUO92002-100RXN; DUO92004-100RXN |

| Reagent or Resource | Source | Identifier |
|---|---|---|
| Effectene Transfection Reagent | QIAGEN | Cat# 301427 |
| BCA Protein Quantification Kit | Vazyme | Cat# E112-01 |
| Pierce™ Protein A/G Magnetic Beads | Thermo Fisher Scientific | Cat# 88802 |
| FastPure Plasmid Mini Kit | Vazyme | Cat# R211-01 |
| Fetal bovine Serum | Gibco | Cat#C11875500CP |
| Penicillin and streptomycin | Thermo Fisher Scientific | Cat#15140163 |
| Taq Pro Universal SYBR qPCR Master Mix | Vazyme | Cat# Q712-02 |
| HiScript II 1st Strand cDNA Synthesis Kit | Vazyme | Cat# R211-01 |
| Taq Pro Universal SYBR qPCR Master Mix | Vazyme | Cat# Q712-02 |
| Phanta Max Master Mix | Vazyme | Cat# P515 |
| Super Signal West Pico PLUS | Thermo | Cat# 34577 |
| Schneider's *Drosophila* Medium | Thermo Fisher | Cat# 21720024 |
| DMEM | Thermo Fisher | Cat# C11330500BT |
| Normal Goat Serum | Solarbio | Cat# SL038 |
| Anti-HA Magnetic Beads | MedChemExpress | Cat# HY-K0201 |
| Anti-Flag Magnetic Beads | MedChemExpress | Cat# HY-K0207 |
| DAPI | Beyotime | Cat# C1002 |
| Trizol | Thermo Fisher | Cat# 15596026 |
| CellTiter-Glo Luminescent Cell Viability Assay Kit | Promega | Cat #G7572 |
| Lentivirus concentration kit | Genome DiTech | Cat #GM-040801-100 |
| pLKO.TRC.1 (plasmid) | Addgene | RRID: ddgene_10878 |
| pLV-IRES-ZsGreen1 (plasmid) | Gift from Fei Lu | NA |
| pUAST (plasmid) | Gift from Xiaowei Guo | NA |
| pMD2.G (plasmid) | Addgene | RRID: ddgene_12259 |
| psPAX2 (plasmid) | Addgene | RRID: ddgene_12260 |
| TrypLE expressed enzyme | Thermo Fisher Scientific | Cat #12604021 |
| Polyethylenimine | Polyscience | Cat #23911-1 |
| neurobasal medium | Gibco | Cat# 12349-015 |
| B27 supplement | Gibco | Cat# 11360-070 |
| penicillin/streptomycin | Invitrogen | Cat# SV30010 |
| basic human fibroblast growth factor | R&D systems | Cat# 4114-TC |

| Reagent or Resource | Source | Identifier |
|---|---|---|
| human epidermal growth factor | R&D systems | Cat# 236-EG |
| GlutaMax Supplement | Gibco | Cat# 35050-061 |
| Sodium pyruvate | Gibco | Cat# 11360-070 |
| fetal bovine serum | CellMax | Cat# SA211.02 |
| ᴅ-Luciferin potassium salt | Beyotime | Cat# ST196 |
| **Oligonucleotides** | | |
| qPCR for-XIAP-F | ACCGTGCGGTGCTTTAGTT | N/A |
| qPCR for-XIAP-R | TGCGTGGCACTATTTTCAAGATA | N/A |
| qPCR for-CYR61-F | CTCGCCTTAGTCGTCACCC | N/A |
| qPCR for-CYR61-R | CGCCGAAGTTGCATTCCAG | N/A |
| qPCR for-AXL-F | GTGGGCAACCCAGGGAATATC | N/A |
| qPCR for-AXL-R | GTACTGTCCCGTGTCGGAAAG | N/A |
| qPCR for-AREG-F | GTGGTGCTGTCGCTCTTGATA | N/A |
| qPCR for-AREG-R | CCCCAGAAAATGGTTCACGCT | N/A |
| qPCR for-CCDN1-F | GCTGCGAAGTGGAAACCATC | N/A |
| qPCR for-CCDN1-R | CCTCCTTCTGCACACATTTGAA | N/A |
| qPCR for-HERPUD1-F | ATGGAGTCCGAGACCGAAC | N/A |
| qPCR for-HERPUD1-R | TTGGTGATCCAACAACAGCTT | N/A |
| qPCR for-APC-F | AAAATGTCCCTCCGTTCTTATGG | N/A |
| qPCR for-APC-R | CTGAAGTTGAGCGTAATACCAGT | N/A |
| qPCR for-HES1 -F | TCAACACGACACCGGATAAAC | N/A |
| qPCR for-HES1 -R | GCCGCGAGCTATCTTTCTTCA | N/A |
| qPCR for-TCF7-F | CTGGCTTCTACTCCCTGACCT | N/A |
| qPCR for-TCF7-R | ACCAGAACCTAGCATCAAGGA | N/A |
| qPCR for-MYC-F | GGCTCCTGGCAAAAGGTCA | N/A |
| qPCR for-MYC-R | CTGCGTAGTTGTGCTGATGT | N/A |
| qPCR for 18S-human F | GGCCCTGTAATTGGAATGAGTC | N/A |
| qPCR for 18S-human R | CCAAGATCCAACTACGAGCTT | N/A |
| qPCR for GAPDH-human F | GGAGCGAGATCCCTCCAAAAT | N/A |
| qPCR for GAPDH-human R | GGCTGTTGTCATACTTCTCATGG | N/A |
| shNR1D2#1 | CCAATGAGTAAGTCTCCATAT | N/A |
| shNR1D2#2 | CCAGTACAAGAAGTGCCTGAA | N/A |
| shNT | CAACAAGATGAAGAGCACCAA | N/A |
| qPCR for CUT&Tag-Axl-F | CGCCATGATACCTGGTTAAG | N/A |
| qPCR for CUT&Tag-Axl-R | CGCTTGAGCCCAGGAGTTC | N/A |
| qPCR for CUT&Tag-Hey1-F | GCCTCGGTCCCAGAAATCA | N/A |
| qPCR for CUT&Tag-Hey1-R | CGTCCACAGTGGCCACCA | N/A |
| qPCR for CUT&Tag-TCF-F | GAACTTACTACAGTCAAAGCAGCT | N/A |
| qPCR for CUT&Tag-TCF-R | CCGCTTTGGTCTTGATCTTG | N/A |
| DNA Spike in F | GCCTTCTTCCCATTTCTGATCC | N/A |

| Reagent or Resource | Source | Identifier |
|---|---|---|
| DNA Spike in R | CACGAATCAGCGGTAAAGGT | N/A |
| PCR for E75-F | ATGGTTTGTGCAATGCAAGA | N/A |
| PCR for E75-R | TTACGCCTCCAGCATTACCTT | N/A |
| PCR for NR1D2-F | ATGGAGGTGAATGCAGGAGG | N/A |
| PCR for NR1D2-R | TTAAGGGTGAACTTTAAAGGCCA | N/A |
| **Software and algorithms** | | |
| Prism 8.0 | GraphPad Software | RRID:SCR_002798 |
| ImageJ | https://imagej.net/ | RRID:SCR_003070 |
| GEPIA2 | http://gepia2.cancer-pku.cn/#index | NA |
| NIS-Element Viewer | https://www.microscope.healthcare.nikon.com/ | NA |
| BioRender | https://www.biorender.com | NA |
| **Deposited data** | | |
| RNA-seq and CUT&Tag-seq raw data NCBI accession number | https://www.ncbi.nlm.nih.gov/bioproject/1067005 | RNA-seq and CUT&Tag-seq raw data NCBI accession number |
| **Experimental models: Cell lines** | | |
| *Drosophila* S2 cells | Gift from José C. Pastor-Pareja (Tsinghua University) | FBtc9000001 RRID: CVCL_TZ72 |
| U87 MG | American Type Cell Collection (ATCC) | HTB-14 RRID: CVCL_0022 |
| MGG6 | Gift from Jeremy Rich (University of Pittsburgh) | RRID: CVCL_D1H2 |
| MGG4 | Gift from Jeremy Rich (University of Pittsburgh) | RRID: CVCL_D1H1 |
| HEK293T | American Type Cell Collection (ATCC) | CRL-3216 RRID: CVCL_0063 |

## Methods and protocols

### Fly husbandry and genetics

*Drosophila* stocks and crosses were maintained on standard food at a temperature of 25 °C unless otherwise specified. A comprehensive list of all *Drosophila* lines utilized can be found in the reagents and tools table, and detailed genotypes for each figure panel are provided in Dataset EV1. The standard food composition consisted of a cornmeal-yeast mixture, comprising 50 g of corn flour, 30 g of brown sugar, 24.5 g of dry yeast, 7.25 g of white sugar, 9 g of agar, 4.4 mL of propionic acid, 12.5 mL of ethanol, and 1.25 g of nipagin per liter.

### Co-IP and Western blot

Tumor samples or cells expressing indicated constructs with corresponding genotypes were collected and lysed in NP40 buffer with PMSF. Subsequently, the resulting cell lysate was combined with pre-washed magnetic beads, and the mixture was subjected to gentle rotation at room temperature for 30 min. This allowed for efficient binding of the target proteins to the magnetic beads. A magnetic separation rack was utilized, facilitating quick and easy isolation of the beads. The pre-cleared lysate, devoid of magnetic beads, was then carefully transferred to a clean tube, ensuring the retention of the protein content for subsequent analysis. Then, add primary antibody or HA-conjugated beads to 200 µl cell lysate, and incubate overnight at 4 °C. After that, the pre-washed magnetic beads were added to the lysate and antibody solution (omit this step

for antibody-conjugated beads), the bound protein can be finally separated through the magnetic beads. Finally, the coprecipitated protein was eluted for western blot analysis. The proteins were separated using SDS-PAGE and then transferred onto PVDF membranes. To prevent non-specific binding, the membranes were blotted with 5% skim milk powder in TBST for 60 min. Following this, the membranes were incubated overnight at 4 °C with primary antibodies. The following antibodies were used: HA (CST, 1:1000), Flag (Proteintech, 1:1000), Myc (CST, 1:1000), NICD (DSHB, 1:1000), Yki (1:1000), TEAD (CST, 1:500), RBPJ (CST, 1:500), TAZ (CST, 1:500), and NICD1(CST, 1:500). The membranes were washed three times with TBST to remove any unbound antibodies. Afterward, the membranes were incubated with HRP-conjugated secondary antibodies to enable visualization of the target proteins. The following antibodies for western blotting were used: primary antibodies, rabbit anti-Flag (1:6,000), and secondary antibody anti-rabbit-HRP (1:8,000).

### Duolink in situ proximity ligation assay (PLA)

Wing discs and cultured cells were harvested and subsequently fixed. Following fixation, the samples were blocked to prevent non-specific binding and then subjected to a series of washes. The indicated primary antibodies from different origins (mouse or rabbit) were applied and allowed to incubate with the samples. Primary antibodies used *Drosophila* tissues include mouse anti-Flag (Sigma, 1:200) and rabbit anti-HA (CST, 1:200), rabbit anti-Flag (CST, 1:200) and mouse anti-Su(H) (Santa Cruz, 1:100), mouse anti-NICD (DSHB, 1:100), rabbit anti-Sd (from Lei Zhang, 1:100), rabbit anti-Yki (from Duojia Pan, 1:100). Primary antibodies used in mammalian cells include mouse anti-NR1D2 (Santa Cruz, 1:100), rabbit anti-TAZ, rabbit anti-pan TEAD (CST, 1:200), rabbit anti-Yap1 (CST, 1:200), rabbit anti-RBPJ (CST, 1:200) and rabbit anti-NICD1 (CST, 1:200). To enable detection, secondary antibodies conjugated to complementary PLA (Proximity Ligation Assay) probes were added. The Duolink method was employed, involving hybridization, ligation, amplification, and detection steps, following the guidelines provided by the manufacturer. This ensured accurate and reliable detection of protein-protein interactions or protein localization. For control samples, primary antibodies were omitted, and only secondary antibodies were added, which were subsequently detected using standard methods.

### Cell proliferation assay

A total of 2500 cells were plated in each well of the 96-well plates. Subsequently, the Cell Titer-Glo Luminescent Cell Viability Assay Kit (Promega, Cat #G7572) was employed to assess cell viability at the indicated time points following the manufacturer's instructions.

### Mouse experiments

All mouse experiments were conducted in accordance with the relevant guidelines and under an animal protocol (#XQ-19-028) approved by the Institutional Animal Care and Use Committee of Westlake University. The intracranial transplantation of GSCs followed a previously described method (Liu et al, 2023). Briefly, GSC spheres were dissociated into single cells with TrypLE expressed enzyme (Thermo Fisher Scientific, Cat#12604021), and 10,000 cells were injected into the right cerebral cortex of NSG (NOD.Cg-Prkdc scid Il2rg tm1Wjl /SzJ) immunocompromised mice (Shanghai Model Organisms Center, Inc.) individually. As to

in vivo tumor growth comparison, mice were narcotized and then imaged by bioluminescence imaging at the indicated time. In parallel survival experiments, mice were monitored until the development of neurological signs or morbidity symptoms.

### shRNA Plasmids

The shRNA sequences were inserted into the pLKO.TRC.1 plasmid (Addgene, Cat#10878, RRID: Addgene_10878).

### Lentivirus production

HEK293T cells were co-transfected with a lentiviral expression vector, the envelope plasmid pMD2.G (Addgene, Cat#12259, RRID: Addgene_12259), and the packaging plasmid psPAX2 (Addgene, Cat#12260, RRID: Addgene_12260) using polyethylenimine (PEI) (Polyscience, Cat # 23966-1) according to the manufacturer's instructions. After 48 h of transfection, lentiviral particles were harvested and concentrated using the lentivirus concentration kit (Genomeditech, Cat# GM-040801-100).

### Immunofluorescence and imaging

The imaginal discs were dissected from the third-instar larvae on the 6th day after egg laying unless otherwise indicated. Discs were dissected in cold PBS and fixed with 4% paraformaldehyde for 15 min at room temperature, then washed for $3 \times 5$ min with PBS containing 0.1% Triton X-100 solution (PBST). Samples were blocked in 10% goat serum in PBST for 30 min after fixed and washed, and then incubated with primary antibody at 4 °C overnight. Primary antibodies used include mouse Anti-beta Galactosidase (1:100, DSHB), mouse anti-Br (1:100, DSHB), rat anti-Elav (1:100, DSHB), mouse anti-Mmp-1 (1:100, DSHB), mouse anti-wg (1:100, DSHB), Cyclin E (1:100, Santa Cruz), rabbit anti-Dcp1 (CST, 1:100), mouse anti-dMyc (1:50, DSHB), mouse anti-Cut (1:100, DSHB), mouse anti-NICD (1:100, DSHB). After that, samples were washed $3 \times 10$ min with PBST and incubated with secondary antibody and DAPI at 1:200 in PBST for 2 h. The images were performed with the Nikon A1R confocal Microscope. Images were processed with NIS-Element Viewer and ImageJ software.

### Total RNA extraction and quantitative RT-PCR

In this study, a total of 1 million GSCs or a total of 100 eye-antenna-disc tumors or a total of 150 wing imaginal discs were harvested at the late stage of third-instar larvae for total RNA extraction of each biological replicate across different genotypes by using Trizol (Invitrogen, Carlsbad, CA) according to the manufacturer's instructions. The RNA quality examination was conducted as follows: the purity of the sample was determined by NanoPhotometer® (IMPLEN, CA, USA), while the concentration and integrity of RNA samples were detected by using an Agilent 2100 RNA nano 6000 assay kit (Agilent Technologies, CA, USA). Subsequently, the extracted RNA was reverse-transcribed into complementary DNA (cDNA) using a cDNA reverse transcription kit (Vazyme). Taq Pro Universal SYBR qPCR Master Mix (Vazyme) and the corresponding primers were utilized for quantitative polymerase chain reaction (qPCR), and the qPCR reactions were performed on a Jena Qtower384G Real-Time PCR System. 18S and GAPDH were used as the internal control in mammalian cells, and Spike-in served as an internal control for CUT&Tag qPCR.

### Library preparation for RNA sequencing

For each sample, a total amount of 1–3 μg RNA was used as the input for library preparation by strictly following the standard protocol of VAHTS Universal V6 RNA-seq Library Prep Kit for Illumina® (NR604-01/02). Briefly, mRNA was purified by using poly-T oligo-attached magnetic beads from total RNA. The short fragments of mRNA were obtained by adding the fragmentation buffer. After the first strand of cDNA was synthesized by using random hexamer primer and RNase H, the second strand synthesis was performed subsequently by using buffer, dNTPs, DNA polymerase I, and RNase H. And then, the double-stranded cDNA was purified by using QiaQuick PCR kit or AMPure P beads. The purified products of each sample were repaired at the end, added tail, and connected to the sequencing connector, then the appropriate fragment size was selected, and the final cDNA library was obtained by PCR amplification for further sequencing performed on Illumina NovaSeq 6000 platform with NovaSeq 6000 S4 Reagent kit V1.5.

### RNA-seq data processing and analysis

Both wing disc and eye-antennal disc transcriptomic data were processed and analyzed as follows: Data quality control and reads statistics were performed by using FastQC (v0.11.8) software (https://github.com/s-andrews/FastQC/releases/tag/v0.11.8). Low-quality reads were removed, and the maintained high-quality reads (Q30 > 90%) were mapped to the Ensemble (Martin et al, 2023) *Drosophila melanogaster* reference genome (Drosophila_melanogaster.BDGP6.32.108) by using Hisat2 (Kim et al, 2019). HTSeq (Anders et al, 2015) was applied for gene feature counting with default settings. Differential gene expression identification and functional annotation were conducted in R (v4.2.0). Genes meeting |fold-change| >1.5 and false discovery rate (FDR) <0.05 were identified as differentially expressed genes (DEGs) by using edgeR (Robinson et al, 2010) with the "RLE" method. The gene ID transformation was performed by using biomaRt (Durinck et al, 2009). DEGs were annotated against terms in Kyoto Encyclopedia of Genes and Genomes (KEGG) database (Kanehisa et al, 2017; Kanehisa and Goto, 2000) and Gene Ontology (GO) consortium (Ashburner et al, 2000) and by using clusterProfiler package (Yu et al, 2012). Data visualization was relied on pheatmap (v1.0.12), RColorBrewer (v1.1-3), and ggplot2 (v3.4.4) packages. Gene set enrichment analysis (GSEA) (Mootha et al, 2003; Subramanian et al, 2005) was performed locally with a gene list of each pathway from FlyBase (https://flybase.org) (Gramates et al, 2022).

### Cleavage under targets and tagmentation (CUT&Tag)

All CUT&Tag experiments were conducted by strictly following the manual instructions of the Hyperactive Universal CUT&Tag Assay Kit for Illumina Pro (Vazyme, TD904). Briefly, ~$1 \times 10^5$ cells were collected from wing imaginal disc or eye-antennal disc tumors for each sample group (at least three biological replicates). These cells were incubated and mixed (2–3 times) with Concanavalin A beads Pro at room temperature for 10 min. Then, the liquid was removed, and the specific primary antibodies were added with ice-cold 50 μl Antibody Buffer. The primary antibodies used include Myc-tag (CST, 1:100), NICD (DSHB, 1:50), Flag-tag (Sigma, 1:50) in *Drosophila* tissues; Rabbit IgG (Beyotime, 1:100) or HA-tag (CST, 1:100) for CUT&Tag qPCR in mammalian cells. After incubating overnight at 4 °C, the liquid was removed, and the secondary antibody diluted by Dig-Wash Buffer (1:100) was added for incubation at room temperature for 1 h. For each sample, 2 μl Hyperactive pA/G-Transposon Pro mixed with 98 μl Dig-300 Buffer were added and incubated at room temperature for 1 h. After

gently washing with Dig-300 Buffer, the fragmentation of each sample was conducted by incubating the mixture of 40 μl Dig-300 Buffer, 10 μl 5 × TTBL, and the products obtained in the previous step at 37 °C for 1 h. About 1 pg spike-in was added for internal control before DNA extraction. Then, DNA extraction was performed by incubating the products with DNA Extract Beads Pro diluting in 50 μl 2 × B&W Buffer at room temperature for 20 min. The i7 and i5 Indexed Primer were combined for PCR amplification with the recommended cycle number (9–11). Finally, only the CUT&Tag libraries passing fragment analyzer quality control (Fragment Analyzer-12/96) were then sequenced on Illumina NovaSeq 6000.

### CUT&Tag-Seq data processing and analysis

The analytic procedures of CUT&Tag-seq data were referred to recommended pipelines (Henikoff et al, 2020). The raw sequencing data were processed by using FastQC (v0.11.8) software (https://github.com/s-andrews/FastQC/releases/tag/v0.11.8) for quality control. Adapters were removed and reads were trimmed by using Cutadapt (v1.18) (Martin, 2011). The clean reads were first mapped to the Ensemble (Martin et al, 2023) *Escherichia coli* (*E. coli*) reference genome (GCF_000005845.2_ASM584v2_genomic) by using Bowtie2 (v2.4.2) (Langmead and Salzberg, 2012), the unrecognized reads were subsequently mapped to *Drosophila melanogaster* reference genome (Drosophila_melanogaster.BDGP6.32.108) with the parameters "--end-to-end --very-sensitive --no-mixed --no-discordant --phred33 -I 0 -X 1000 --no-unal". After using SAMtools (v1.11) (Danecek et al, 2021) to transform SAM files into BAM files, Picard (v2.25.1) (https://broadinstitute.github.io/picard/) was applied to remove duplicates, and the reads were selectively maintained when meeting "mapping_quality ≥20". The final BAM files of the same genotype among the replicates were merged into one for peak calling, which is performed by using model-based analysis of ChIP-seq (MACS2, v2.2.6) (Zhang et al, 2008b) with settings "-g dm -f BAMPE -q 0.05 --keep-dup all". Peaks were annotated by using the ChIPseeker (Yu et al, 2015) package in R (v4.2.0), and the peak distribution was visualized by using *plotAnnoBar* and *plotDistToTSS* functions. The final BAM files were transformed into bigwig files for signal enrichment visualization by using the *bamCoverage* function of BEDTools (v2.30.0) (Quinlan and Hall, 2010) with settings "--binSize 20 --normalizeUsing BPM". The average normalized signal values of peaks in the *Drosophila* genome region were calculated and visualized by using DeepTools (v3.5.1) (Ramírez et al, 2016), in which regions within 3 kb distance relative to the transcriptional start sites (TSS) were included, and gene body region was scaled into 5 kb length. Integrative genomics viewer (IGV) (Robinson et al, 2011) was applied to visualize the peaks on specific genome regions of interested genes. The binding motifs were identified by using the *findMotifsGenome* or *findMotifs* functions of Homer (v4.11) (Heinz et al, 2010) regarding to conditions. The motif scanning algorithm, Find Individual Motif Occurrences (FIMO, v5.5.5) (Grant et al, 2011) was applied in a set of sequences to determine all the positions where transcription factor motifs match ($p < 0.0001$) via The MEME Suite (Bailey et al, 2015).

### Public data analysis

The binding regions of Yki and Su(H) on *Drosophila melanogaster* genome were annotated by using public datasets GSE38594 (Oh et al, 2013) and GSE41429 (Djiane et al, 2013), respectively. The DEG list between siNR1D2 and siControl LN-18 cells was retrieved from the Supplementary Table S1 and S2 of our previous publication(Yu et al, 2018), and the over-representation analyses (ORA) were conducted using clusterProfiler package (Yu et al, 2012). The analysis of *NR1D2* relative expression level between GSC and NSC was conducted by using the processed data (CPM) from GSE54791 (Mack et al, 2019) for the Mann–Whitney test. The correlation analysis of *NR1D2* expression with other genes of interest were performed by using the GEPIA2 web server (Tang et al, 2019).

### Quantification and statistical analysis

In Figs. 1F–H, 3B–D,F and EV1D–H,L, EV3G–H,K,L the region of interest (ROI) in each clone and the wild-type cell was subjected to cycling and measurements using ImageJ. The size of the tumors or ROI was quantified by measuring the area of the GFP-positive region in each sample. All statistical analyses were performed with GraphPad Prism 8.0. software. Data represents mean values ± SD. Statistical significance was assessed using appropriate methods depending on the experimental design. For comparisons between the two groups, an unpaired two-tailed Student's *t*-test was performed. For experiments involving three or more groups, ordinary one-way ANOVA or two-way ANOVA analysis was conducted. Additionally, in survival analysis, the log-rank (Mantel-Cox) test was employed. The investigators analyzing the data were not blind to the identity of the samples. The specific statistical tests used for each analysis are indicated in the corresponding figures. $p$ value $<0.05$ was considered significant, $^{*}p < 0.05$, $^{**}p < 0.01$, $^{***}p < 0.001$, $^{****}p < 0.0001$.

## Data availability

All sequencing data of this study is deposited in the National Center for Biotechnology Information Sequence Read Archive (Rubin et al, 2000) with the accession number BioProject: PRJNA1067005.

The source data of this paper are collected in the following database record: biostudies:S-SCDT-10_1038-S44318-024-00290-3.

## Peer review information

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

## Acknowledgements

We thank Jiong Chen, Oren Schuldiner, Tian Xu, Lei Xue, Duojia Pan, Suning Liu, Sheng Li, Zizhang Zhou, Zheng Guo, Hai Huang, Lei Zhang, Bloomington *Drosophila* Stock Center, Vienna *Drosophila* Resource Center, and Developmental Studies Hybridoma Bank for providing fly stocks and reagents; the Microscopy Core Facility, the Genomics Core Facility, the General Equipment and Autoclave Service Core Facility, and the High-Performance Computing Center of Westlake University for the facility support and technical assistance; Wenhan Liu for fly stock maintenance. Cartoons were created with BioRender.com. This project was supported by the National Natural Science Foundation of China (32170824 and 32322027) to X.M. and (82073268) to Q.X., HRHI program (1011103360222B1) of Westlake Laboratory of Life Sciences and Biomedicine to X.M., Westlake Education Foundation to Q.X., "Pioneer" and "Leading Goose" R&D Program of Zhejiang (2024SSYS0034 and 2024SSYS0036).

## Author contributions

**Xianping Wang**: Validation; Investigation; Writing—original draft; Project administration; Writing—review and editing; Xianping Wang performed majority of fly related experiments, Co-IP, RNA-seq and CUT&Tag experiment. **Yifan Guo**: Formal analysis; Validation; Investigation; Visualization; Writing—original draft; Yifan Guo analyzed the bulk RNA-seq and CUT&Tag data. **Peng Lin**: Validation; Investigation; Visualization; Peng Lin performed the GBM-related experiments in Figs. 6A–D,I,J,K-K' and EV5D,H-H',I and analyzed the data. **Min Yu**: Resources; Validation; Investigation; Visualization; Min Yu provided the NR1D2 plasmid used in Fig. 6F,G and performed other unpublished data related to Fig. 6. **Sha Song**: Validation; Investigation; Visualization; Sha Song helped to perform the CUT&Tag-qPCR experiment in Fig. 6H, and provided NICD, Su(H) and RBPJ plasmid used in the preliminary experiment related to Figs. 5F,G, 6G. **Wenyan Xu**: Validation; Investigation; Visualization; Wenyan Xu helped to perform the Co-IP experiments in Figs. 5D–G, 6F–H. **Du Kong**: Resources; Do Kong provided the *Scrib^{−/−},wts^{−/−}* double mutant fly used in the paper. **Yin Wang**: Investigation; Yin Wang helped to perform a partial CUT&Tag experiment in Fig. 3I,G,H. **Yanxiao Zhang**: Supervision; Validation; Investigation; Yanxiao Zhang helped to design and analyze CUT&Tag data in Figs. 5J and EV4E–H. **Fei Lu**: Supervision; Funding acquisition; Validation; Project administration; Writing—review and editing; Fei Lu provided the U87 cell line used in Fig. 6F–H and designed other unpublished data related to Fig. 6. **Qi Xie**: Supervision; Funding acquisition. **Xianjue Ma**: Conceptualization; Supervision; Funding acquisition; Writing—original draft; Project administration; Writing—review and editing.

Source data underlying figure panels in this paper may have individual authorship assigned. Where available, figure panel/source data authorship is listed in the following database record: biostudies:S-SCDT-10_1038-S44318-024-00290-3.

## Disclosure and competing interests statement

The authors declare no competing interests.

# Expanded View Figures

**Figure EV1.  Ecdysone signaling inhibition promotes tumor malignancy.**

(A) Quantification of the relative size of GFP positive regions in Fig. 1B ($n = 6$, 10, 9, 5, 8, 6). *n* represents the number of biological replicates. Statistical analysis by ordinary one-way ANOVA test; mean ± SD. ***$p = 0.0004$, *$p = 0.0378$. (B) Volcano plot of differentially expressed genes (DEGs) in *scrib*$^{-/-}$,*wts*$^{-/-}$ and *scrib*$^{-/-}$,*Ras*$^{v12}$ tumors (top panel). Log2 fold-change of *E75* expression in *scrib*$^{-/-}$,*Ras*$^{v12}$ vs *scrib*$^{-/-}$,*wts*$^{-/-}$ tumors (bottom panel). (C) GSEA enrichment of ecdysone-related genes in *scrib*$^{-/-}$,*wts*$^{-/-}$ and *scrib*$^{-/-}$,*Ras*$^{v12}$ tumors. The FWER $p < 0.05$ served as the significance threshold. (D) Confocal images of eye-antennal discs bearing *ey-Flp*-MARCM-induced mosaics of each genotype stained with Broad (Br) antibody. (D') Quantification of relative Br intensity of GFP positive mosaics clones ($n = 6$, 7, 8). *n* represents the number of biological replicates. Statistical analysis by ordinary one-way ANOVA test; mean + SD. ****$p = 0.0000000006$, ***$p = 0.00045$. (E) Dorsal views of *ey-Flp*-MARCM-induced GFP-positive tumor-bearing larvae and the corresponding dissected tumor (right). Quantification of pupation rate of tumor-bearing larvae (E') and relative tumor size of GFP positive mosaics clones (E", $n = 9$, 4, 8, 6). *n* represents the number of biological replicates. (E") Statistical analysis by ordinary one-way ANOVA test; mean ± SD. *$p = 0.0222$, ****$p = 0.000095$, **$p = 0.0014$. (F, G) Confocal images of eye-antennal discs bearing *ey-Flp*-MARCM-induced mosaics of each genotype. (F', G') Quantification of the relative size of GFP positive regions (F', $n = 8$, 5, 6, 5; G', $n = 9$, 9). *n* represents the number of biological replicates. (F') Statistical analysis by ordinary one-way ANOVA test; mean ± SD. ns non-significant, $p = 0.1748$. (G') Statistical analysis by students' *t*-test; mean ± SD. ***$p = 0.0001$. (H) Confocal images of *ey-Flp*-MARCM-induced tumor of each genotype stained with Broad (Br) antibody. (H') Quantification of relative Br intensity of GFP positive mosaics clones ($n = 7$, 5, 6, 6). *n* represents the number of biological replicates. Statistical analysis by ordinary one-way ANOVA test; mean ± SD. ****$p = 0.000007$, ****$p = 0.0000009$, ****$p = 0.00000069$. (I, J) Confocal images of eye-antennal discs bearing *ey-Flp*-MARCM-induced mosaics of wild-type and E75 overexpression stained with anti-β-galactosidase antibody for the *EcRE-LacZ* staining (I), anti-Br antibody (J). Clones are circled by the white dashed line. (K) Schematic diagram of ecdysone signaling, ecdysone response genes, and tumorigenesis according to Figs. 1H and EV1I,J. (L) Dorsal views of *ey-Flp*-MARCM-induced GFP-positive tumor-bearing larvae and the corresponding eye disc or tumor (right). (L') Quantification of relative tumor size in (L) ($n = 6$, 5, 6, 6, 6, 6). *n* represents the number of biological replicates. Statistical analysis by ordinary one-way ANOVA test; mean ± SD. ****$p = 0.000092$, ****$p = 0.000000000012$. (L") Quantification of larvae pupation rate in (L). Scale bars: 100 μm (D, F–J), 200 μm (E, L).

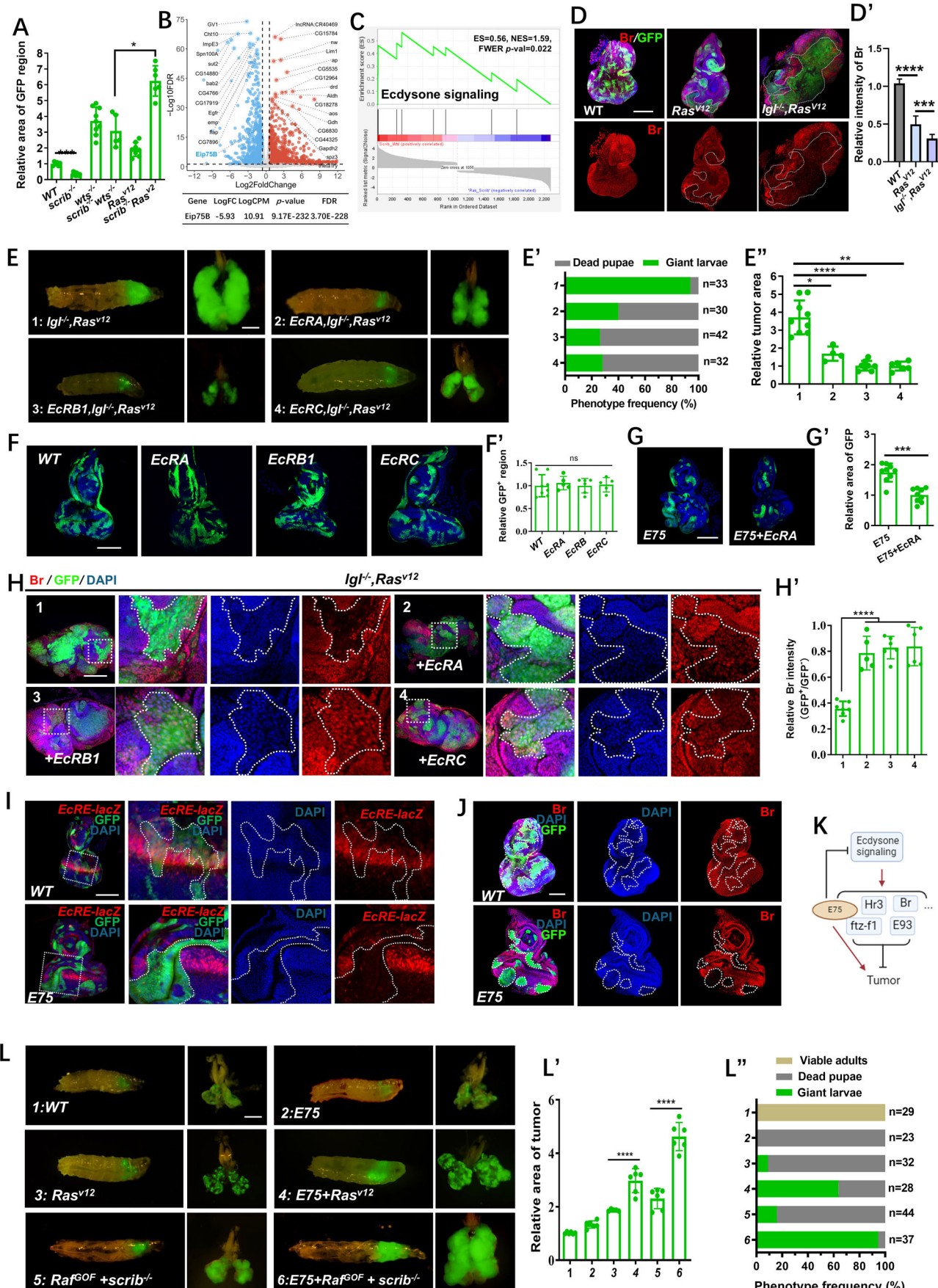

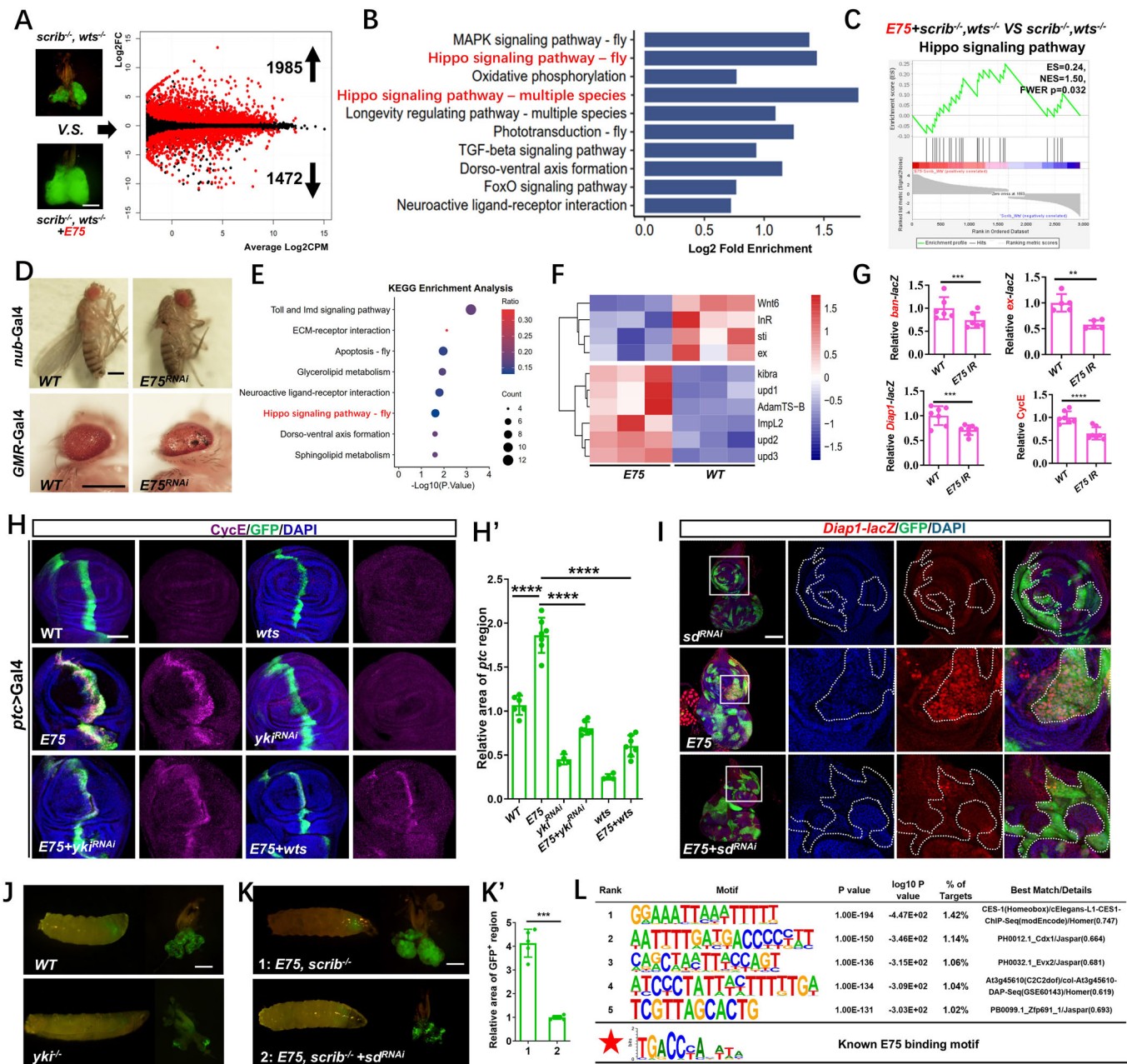

**Figure EV2. E75 overexpression induces Hippo pathway-dependent growth and tumorigenesis.**

(A) MA (Minus-versus-Add) plot of *scrib*$^{-/-}$,*wts*$^{-/-}$ and *E75,scrib*$^{-/-}$,*wts*$^{-/-}$ tumors. (B) Enrichment analysis of signaling pathway in *scrib*$^{-/-}$,*wts*$^{-/-}$ and *E75,scrib*$^{-/-}$,*wts*$^{-/-}$ tumors. (C) GSEA enrichment of Hippo signaling-related genes in *scrib*$^{-/-}$,*wts*$^{-/-}$ and *E75, scrib*$^{-/-}$,*wts*$^{-/-}$ tumors. The FWER $p < 0.05$ served as the significance threshold. (D) Light micrographs of the adult wings (top) and eyes bearing the indicated genotypes. (E) KEGG enrichment of signaling pathways in *scrib*$^{-/-}$,*wts*$^{-/-}$ and *E75, scrib*$^{-/-}$,*wts*$^{-/-}$ tumors. Terms that satisfied the hypergeometric distribution were considered significant ($p < 0.05$). (F) Heatmap profiles of Hippo signaling-related genes in wild-type (WT) and *E75* overexpression wing discs. (G) Quantification of relative staining intensity of GFP positive (E75 knockdown) and negative (wild type) regions for corresponding antibodies in Fig. 3D ($n = 6$, 6 for *ban-lacZ*, $n = 6$, 6 for *ex-lacZ*, $n = 7$, 7 for *Diap1-lacZ*, $n = 7,7$ for CycE). $n$ represents the number of biological replicates. Statistical analysis by paired students' *t*-test; mean ± SD. ***$p = 0.0009$, ***$p = 0.0009$, ***$p = 0.0002$, ****$p = 0.000007$. (H) Confocal image of Cyclin E (CycE) antibody staining of wing discs bearing the indicated genotypes. (H') Quantification of the relative size of *ptc* region of H ($n = 6$, 7, 4, 7, 4, 7). $n$ represents the number of biological replicates. Statistical analysis by ordinary one-way ANOVA test; mean ± SD. ****$p = 0.00000019$, ****$p = 0.0000024$, ****$p = 0.00000086$. (I) Confocal images of *ey-Flp*-MARCM-induced GFP-positive mosaic clones stained with anti-β-galactosidase antibody for the *Diap1-LacZ* staining. (J, K) Dorsal views of *ey-Flp*-MARCM-induced GFP-positive tumor-bearing larvae and the corresponding dissected eye disc or tumor. Quantification of relative tumor size of GFP positive mosaics clones (K', $n = 5$, 6). $n$ represents the number of biological replicates. Statistical analysis by students' *t*-test; mean ± SD. ***$p = 0.004$. (L) HOMER motif analysis of *E75* novel binding motifs in wing pouch region. ZOOPS scoring and hypergeometric distribution test were considered collectively, and motifs with $p < 0.0000001$ were considered significantly enriched. Scale bars: 100 μm (H, I), 200 μm (A, D, J, K).

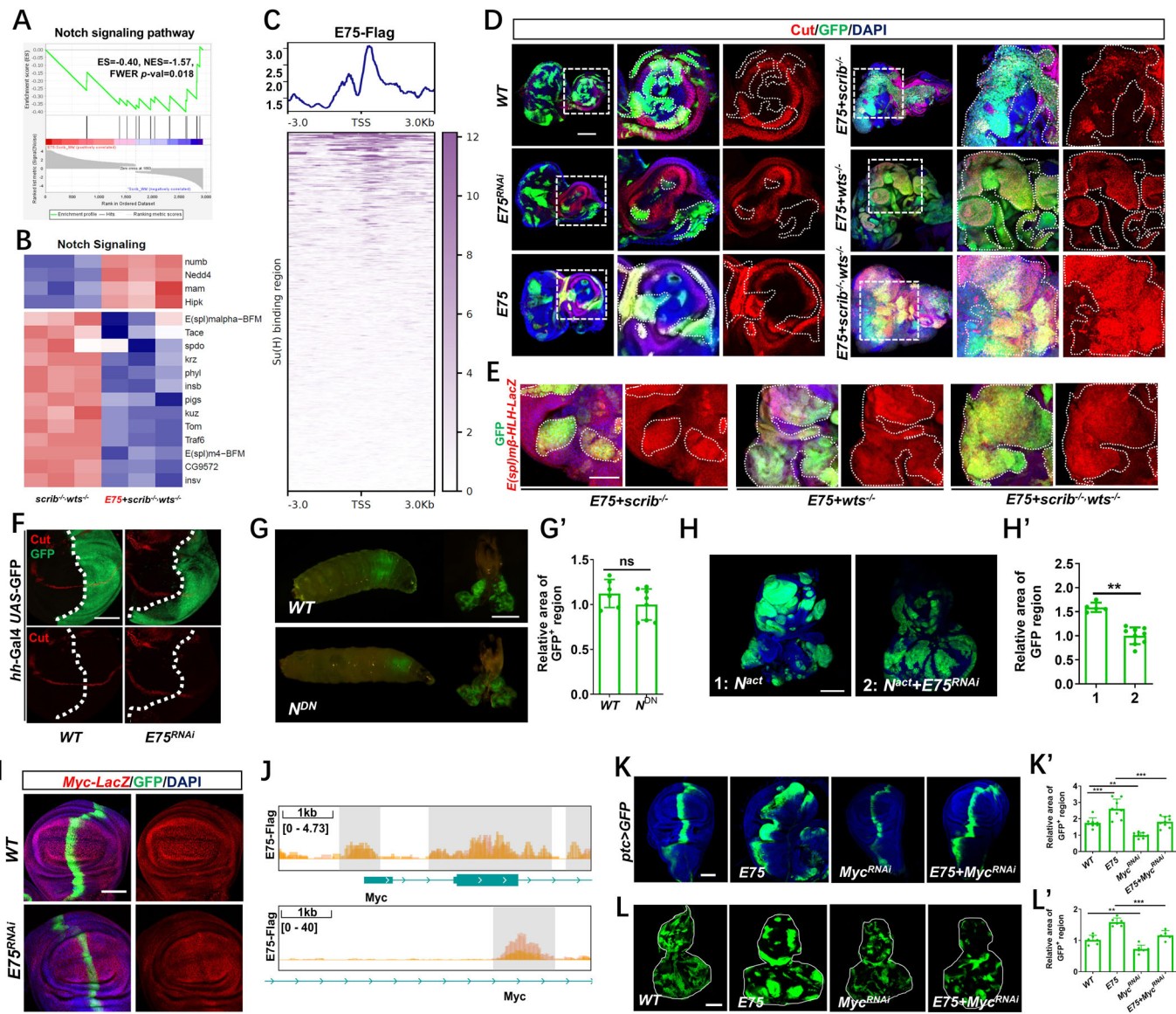

**Figure EV3. E75 positively regulates the Notch pathway.**

(A, B) GSEA enrichment (A) and heatmap profiles (B) of Notch signaling-related genes in $scrib^{-/-}$, $wts^{-/-}$ and $E75$, $scrib^{-/-}$, $wts^{-/-}$ tumors. The FWER $p < 0.05$ served as the significance threshold. (C) Line plots of the average CUT&Tag signal of E75 (top panel) and the heatmaps of the CUT&Tag signals of Su(H) in *Drosophila* (bottom panel). CUT&Tag signals are displayed within a region spanning ±3 kb around all canonical transcription start sites (TSS) genome-wide. (D, E) Representative confocal images of Cut (D) and $E(spl)m\beta$-HLH-lacZ (E) staining in *ey-Flp*-MARCM-induced tumors and clones with indicated genotype. (F) Confocal images of the wing disc of control or with $E75$ knockdown under the control of $hh$ promoter stained with anti-Cut antibody. (G) Dorsal views of *ey-Flp*-MARCM-induced GFP-positive clonal-bearing larvae and the corresponding dissected eye disc. Quantification of the relative size of GFP positive mosaic clones (K', $n = 6, 8$). $n$ represents the number of biological replicates. Statistical analysis by Student's $t$-test; mean ± SD. ns not significant, $p = 0.196$. (H) Confocal images of *ey-Flp*-MARCM-induced GFP-positive mosaic clones and corresponding quantification data (H', $n = 5, 9$). $n$ represents the number of biological replicates. Statistical analysis by Student's $t$-test; mean ± SD. **$p = 0.0044$. (I) Representative confocal images of *Myc-lacZ* staining in wing discs of wild-type and $E75$ knockdown. (J) Browser shots of $E75$ CUT&Tag signal at the regulatory region of *Myc*. (K, L) Representative confocal images of the wing (K) and eye discs (L) with indicated genotypes. (K', L') Quantification of relative GFP region of K and L (K', $n = 8, 8, 9, 9$; L', $n = 6, 6, 7, 5$). $n$ represents the number of biological replicates. Statistical analysis by ordinary one-way ANOVA test; mean ± SD. (K') ***$p = 0.0004$, **$p = 0.0013$, ***$p = 0.00076$. (L') ****$p = 0.0048$, ***$p = 0.0003$. Scale bars: 100 µm (D–F, H, I, K, L), 200 µm (G).

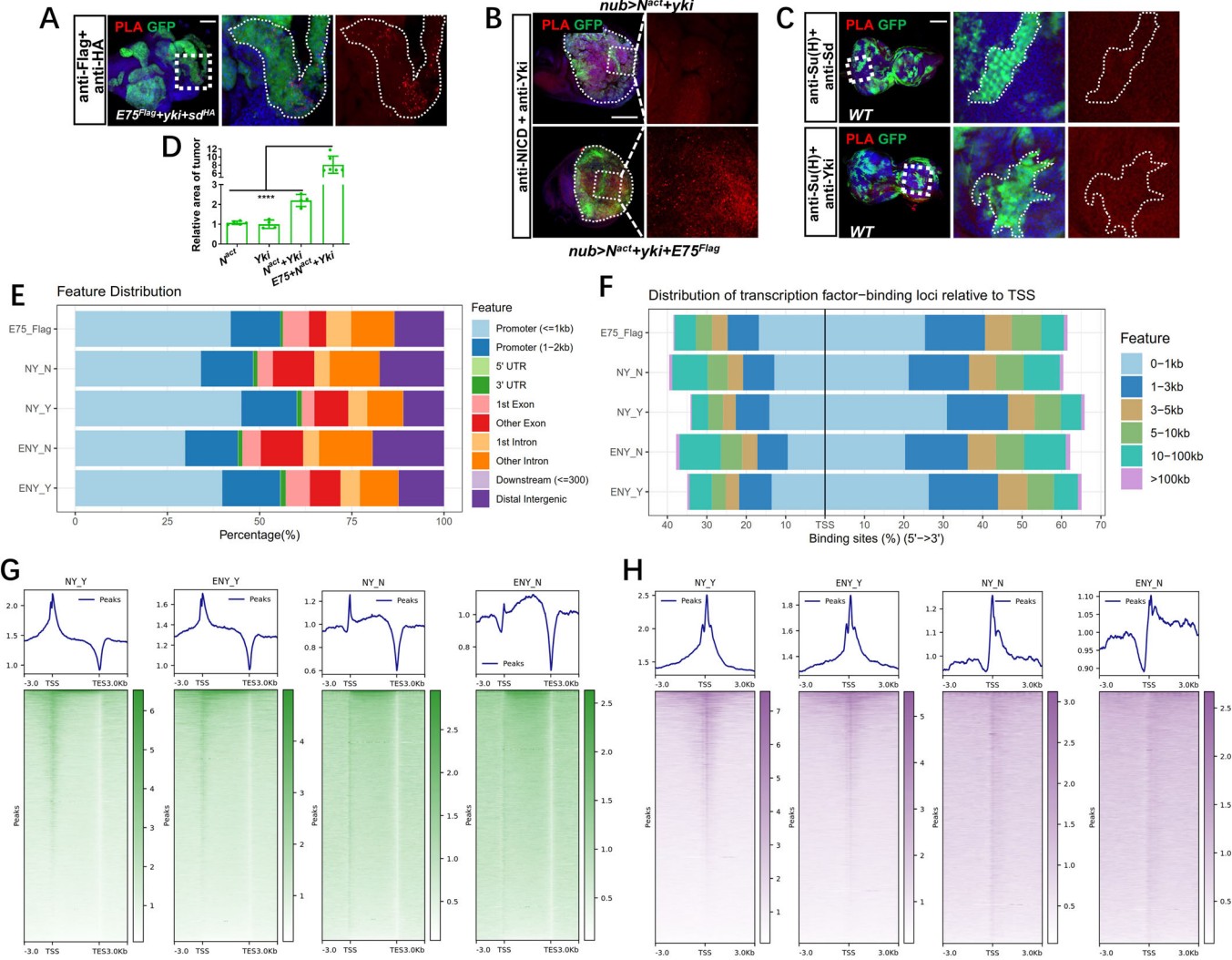

**Figure EV4. E75 regulates the transcription of the Hippo and Notch pathway target genes.**

(A) PLA was performed on eye discs bearing *ey-Flp*-MARCM-induced *E75^Flag^*, *Yki*, and *Sd^HA^* co-expressed clones to test close-proximity interactions between E75 and Sd. (B) PLA was performed on wing discs with *Yki* and *Sd^HA^* co-expression under the control of *nub* promoter, with or without *E75^Flag^* expression, to test close-proximity interactions between NICD and Yki. (C) PLA was performed on eye discs bearing *ey-Flp*-MARCM-induced WT clones to test close-proximity interactions between Su(H) and Sd, as well as Su(H) and Yki. (D) Quantification of relative tumor size of GFP-positive tumor clones in Fig. 5I (*n* = 5, 4, 4, 6). *n* represents the number of biological replicates. Statistical analysis by ordinary one-way ANOVA test; mean ± SD. ****$p$ = 0.0000006, ****$p$ = 0.0000011, ****$p$ = 0.0000112. Error bars from left to right, 0.075, 0.18, 0.26, and 2.54. (E) Feature distribution of genomic annotations of CUT&Tag in E75 peaks (E75_Flag), NICD peaks of *N^act^* and *Yki* overexpression (NY_N), Yki peaks of *N^act^* and *Yki* overexpression (NY_Y). (F) Distribution of binding loci relative to TSS of peaks mentioned in (D). (G) Binding profiles and heatmaps of NICD and Yki in *N^act^*, *Yki* and *E75*, *N^act^*, *Yki* tumors. CUT&Tag signals are displayed within a region spanning −3 kb around all canonical TSS and +3 kb around all canonical transcription end sites (TES). (H) Binding profiles and heatmaps of NICD and Yki in *N^act^*, *Yki* and *E75*, *N^act^*, *Yki* tumors. CUT&Tag signals are displayed within a region spanning −3 kb and +3 kb around all canonical TSS. Scale bars: 100 μm (A–C).

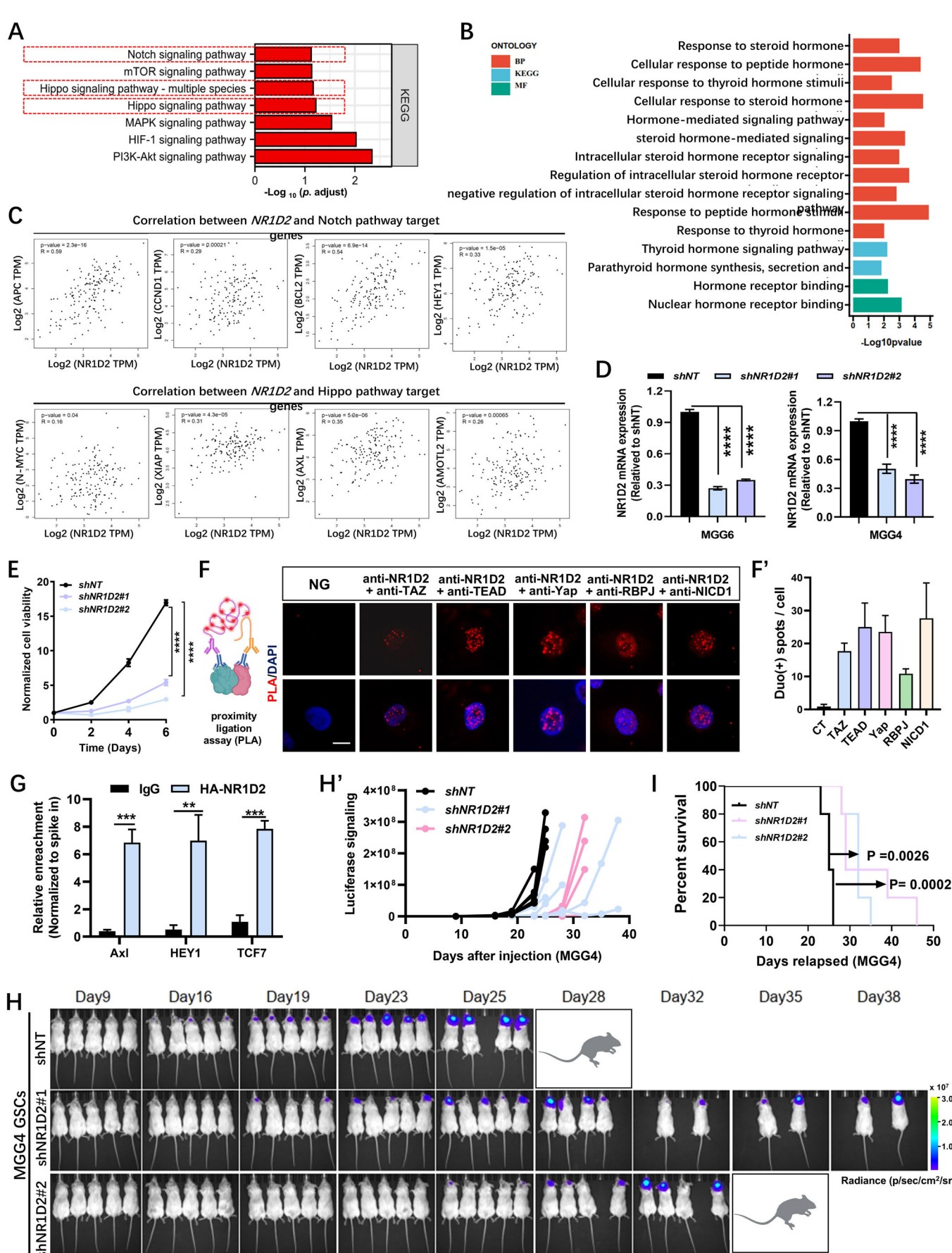

**Figure EV5.  Silencing *NR1D2* suppressed glioblastoma stem cell-driven tumor growth.**

(A) KEGG analysis indicated enrichment of Hippo and Notch signaling in *NR1D2*-depleted U87 MG cells. Terms satisfying the hypergeometric distribution were considered significant ($p < 0.05$). (B) KEGG and GO analyses indicated the enrichment of multiple terms associated with hormone regulation in *NR1D2*-depleted U87 cells. Terms satisfying the hypergeometric distribution were considered significant ($p < 0.05$). (C) Pair-wise gene correlation analysis of *NR1D2* and target genes of Notch pathway (top panels) and Hippo pathway (bottom panels) in GBM by GEPIA2. (D) Relative mRNA expression of *NR1D2* in GSCs and *NR1D2*-depleted cells. $n = 3$ for each sample. *n* represents the number of technical replicates. Statistical analysis by two-tailed Student's *t*-tests; mean + SD. ****$p = 0.0000000022$, ****$p = 0.00000001$, ****$p = 0.000011$, ****$p = 0.0000033$. (E) Relative cell viability of MGG4 of shNT and *shNR1D2*. $n = 3$ for each sample. *n* represents the number of technical replicates. Statistical analysis by two-way repeated measures ANOVA with Dunnett's multiple hypothesis test correction; mean ± SD. ****$p = 0.0000039$, ****$p = 0.000006$. (F) PLA working model (left). PLA analysis in U87 MG cells to test close-proximity interactions between NR1D2 and TAZ, TEAD, Yap, RBPJ, and NICD1 (right). (F') Quantification of PLA signal intensity in (E) ($n = 7, 6, 9, 6, 6$); mean + SD. *n* represents the number of biological replicates. (G) CUT&Tag qPCR analysis of *Axl*, *HEY1*, and *TCF7* in U87 MG. U87 cells transfected with *HA-NR1D2* were used for HA enrichment quantification on promoter region ($-500$ to 0). $n = 3$ for each sample. *n* represents the number of technical replicates. Statistical analysis by two-tailed Student's *t*-tests; mean + SD. ***$p = 0.0003$, **$p = 0.0042$, ***$p = 0.0001$. (H) In vivo bioluminescence imaging of NSG mice bearing tumors on the 9th, 16th, 19th, 23rd, 25th, 28th, 32nd, 35th, and 38th day after MGG4 GSCs were injected into the mice brains. The injected MGG4 GSCs were transfected with *shNT*, *shNR1D2#1*, or *shNR1D2#2*, respectively. (H') Quantification of tumor size by in vivo luciferase assays. (I) Survival curve of NSG mice bearing intracranial tumors from MGG4 GSCs transfected with *shNT*, *shNR1D2#1*, or *shNR1D2#2*, respectively. Statistical analysis by log-rank (Mantel-Cox) test. **$p = 0.0026$, **$p = 0.0002$. Scale bar: 10 μm (F).

