## [Peer Review File · The EMBO Journal]

Nuclear receptor E75/NR1D2 promotes tumor malignant transformation by integrating Hippo and Notch pathways

Xianping Wang, Yifan Guo, Peng Lin, Min Yu, Sha Song, Wenyan Xu, Du Kong, Yin Wang, Yanxiao Zhang, Fei Lu, Qi Xie, and Xianjue Ma

Corresponding authors: Xianjue Ma (maxianjue@westlake.edu.cn) , Qi Xie (xieqi@westlake.edu.cn), Xianping Wang (wangxianping@westlake.edu.cn)

Review Timeline:

Submission Date:	21st Feb 24
Editorial Decision:	8th Apr 24
Revision Received:	1st Jun 24
Editorial Decision:	24th Jul 24
Revision Received:	23rd Aug 24
Editorial Decision:	20th Sep 24
Revision Received:	10th Oct 24
Accepted:	23rd Oct 24

Editor: Ieva Gailite

Transaction Report:

Dear Xianjue,

Thank you for submitting your manuscript for consideration by the EMBO Journal. We have now received comments from two reviewers, which are included below for your information.

Based on the overall interest expressed in the referee reports and your willingness to engage in a major revision as expressed in the preliminary revision plan provided during the pre-decision consultation, I would like to invite you to address the comments of all reviewers in a revised version of the manuscript. I should add that it is The EMBO Journal policy to allow only a single major round of revision and that it is therefore important to resolve the main concerns at this stage.

We generally allow three months as standard revision time, which can be extended if necessary. As a matter of policy, competing manuscripts published during this period will not negatively impact on our assessment of the conceptual advance presented by your study. However, please contact me as soon as possible upon publication of any related work to discuss the appropriate course of action. I have currently extended the deadline to four months as requested. Should you foresee a problem in meeting this deadline, please let me know in advance to discuss an extension.

When preparing your letter of response to the referees' comments, please bear in mind that this will form part of the Review Process File and will therefore be available online to the community. For more details on our Transparent Editorial Process, please visit our website: <https://www.embopress.org/page/journal/14602075/authorguide#transparentprocess>. Please also see the attached instructions for further guidelines on preparation of the revised manuscript.

Please feel free to contact me if you have any further questions regarding the revision. Thank you for the opportunity to consider your work for publication. I look forward to receiving your revised manuscript.

With best regards,

Ieva

Ieva Gailite, PhD
Senior Scientific Editor
The EMBO Journal
Meyershofstrasse 1
D-69117 Heidelberg
Tel: +4962218891309
i.gailite@embojournal.org

We realize that it is difficult to revise to a specific deadline. In the interest of protecting the conceptual advance provided by the work, we recommend a revision within 3 months (7th Jul 2024). Please discuss the revision progress ahead of this time with the editor if you require more time to complete the revisions.

Referee #1:

This manuscript by Wang and colleagues reports on the role of E75, a nuclear receptor and early ecdysone-responsive gene, in tumor progression. The authors found that the level of ecdysone, an insect hormone, was relatively low in malignant tumors. Using genetic tools, they demonstrated that E75 overexpression increases the malignancy of tumors, while knocking down E75 decreases tumor size. Mechanistically, it was shown that E75 can integrate the Hippo and Notch signaling pathways through physical interaction with Hippo and Notch transcription factors, and by co-binding to their targets. Lastly, the authors demonstrate that the function of E75 in regulating Notch and Hippo pathway-mediated tumorigenesis is conserved in mammals. Overall, this work illuminates the role of E75 in integrating the Hippo and Notch pathways during tumor progression.

Major concerns:

1. In the methods section, the authors mentioned that "third-instar larvae imaginal discs" were used. Were the tumors shown in all figures from the same day as the controls? Is the larger tumor size due to a larval developmental delay, resulting in the tumors having extra days to develop? The comparison of tumors in the rescue experiments should also be on the same timeline.
2. Overexpression of E75 could have effects in addition to blocking the EcR signal; it is probably not equivalent to the "hormone inhibition" stated in the abstract. Also, EcR signaling has been shown previously to have a tumor suppressor role in *Drosophila* (Jiang et al., 2018).
3. The majority of experiments involved E75 overexpression, which could artificially affect Cut and Tag results.
4. In mammalian studies, does inhibition of NR1D2 lead to hormonal dysregulation? If so, which hormones are affected?
5. NR1D1 is upregulated in GBM, but E75 is downregulated in fly malignant tumors. This discrepancy should be addressed.
6. In the abstract, the authors mentioned that "malignant epithelial tumors in *Drosophila* exhibit a comparable decrease in the activation of hormone signaling." Does this mean malignant tumors exhibit reduced ecdysone signaling? Or does inhibition of ecdysone signaling promote tumor development into malignant ones?
7. As shown, knockdown of E75 reduces tumor formation in *scrib*^{-/-}, *wts*^{-/-}. What about KD of E75 in *scrib*^{-/-} + Ras tumors? Was tumor size reduced?
8. There are three isoforms of EcR in the fly, differing only at their N termini. The authors showed two in Fig. S1D. What about the third one?
9. S2 cells were used for co-IP assays to examine the physical interactions between E75 and the aforementioned transcription factors (TFs). However, results might differ from those in tumor tissues.
10. In Fig. 2A, the expression of EcRE-lacZ in the E75 OE clone does not show significant changes.
11. Tumor invasive behavior requires Mmp1 activity (PMID: 17301221). In Fig. 2E, E75 OE induces Mmp1 expression in a non-cell autonomous manner? Why migrating tumor cells do not show Mmp1 expression? As Timp was upregulated in the E75 overexpression samples (Fig. 3A), is it possible that Mmp1 expression was inhibited in *scrib*^{-/-}, *wts*^{-/-}+E75 tumors?
12. Fig. 3A, are the samples from the same day? Samples in Fig. 3A contain *wts* mutant clones, is the Hippo inactivation in E75 overexpression due to mutant clone expansion of *wts*^{-/-}?
13. Also in Fig. 3A, since clones were expanded in the *scrib*^{-/-}, *wts*^{-/-}+E75 (Fig. 3d), how come *wts* expression was upregulated in the *scrib*^{-/-}, *wts*^{-/-}+E75 larvae?
14. The conclusion that "E75 negatively regulates Hippo signaling" may not be fully supported by the data presented. The statement that 'Consistently, E75 overexpression or depletion could upregulate or downregulate the expression of multiple target genes in the Hippo pathway, including Wingless (Wg), expanded (ex), Cyclin E (CycE), and bantam (ban) (Fig. 3C)' may need to be revised based on the specific results shown in Figure 3C. Since E75 overexpression did not change the expression of Wg and the expression of Wg, ex, CycE, and ban was not affected in the E75 RNAi samples. It is possible that only high levels of E75 are capable of affecting the transcription of Hippo pathway genes.

Minor comments:

1. Tumor size quantification is required for Fig. 1B.
2. The expression of Cut seems was not reduced in the E75RNAi clone. Also, dotted line in fig. S4D E75RNAi was misplaced.
3. Fig. 4G, Myc-lacZ from E75 overexpression was rotated.
4. Samples for DNA binding profiles analysis were from E75 overexpression? Does E75 overexpression cause additional binding in the genome?
5. Mammals do not produce ecdysone. If the malignancy of fly tumors is due to ecdysone or larval developmental delay, how can the authors conclude that the function of E75 in tumor malignant transformation is conserved in mammals?

Referee #2:

The manuscript of Ma and colleagues unravels a fundamental role of E75 in promoting tumor malignancy in epithelial tumors in *Drosophila* (and in glial tumors in mammals) by positively interacting (genetically and molecularly) with the Notch and Hippo/Yorkie transcriptional machineries. Based on the role of E75 in repressing ecdysone signalling and on the fact that ecdysone signalling is repressed in malignant tumors, authors propose that their experimental model is a paradigm of tumor malignancy caused by hormone depletion. The ms combines the use of *Drosophila* epithelia and a mammalian glial model to demonstrate conservation. The ms is well written, figures self-explanatory and topic very timely. The following issues should be addressed:

Major points:

- Authors claim that the effects of E75 on tumor growth are caused by changes in ecdysone signalling. Authors should validate some of their results (cooperative oncogenesis with *scrib wts*, or others) by depleting ecdysone signalling with other tools.
- I wonder whether the effects of *Ndn* in repressing tumor growth is just a consequence of the deleterious effects of *Ndn* on growth. Some controls would be required. Are *Ndn* expressing clones, cells or tissues affected in growth and survival? I would guess the same happens to *Yki-RNAi* clones where authors might want to use scalloped-RNAi tools to circumvent the effect of *Yki* in "normal" development. Similar comment on *dMyc*. Some controls would be required.
- Difficult to understand that the ability of E75 to induce malignancy of *wts-/-scrib-/-* tumors relies on its capacity to block hippo signalling, since *hpo* is already depleted in *wts-/-* clones. Don't these data indicate that E75 has some other activities besides its role in Hippo signalling. Indeed, Hippo depletion does not lead to malignancy by its own!!!

Other points:

- (1) Changes in the expression levels of *Yki* targets in Figure 3C are not clear. First, *Wg* is not a bona fide *Yki* target. Second, changes are unconvincing in the case of E75 depletion. Some quantification might help.
- (2) Grammar/writing issues should be corrected throughout the ms (some words are lacking, etc). Please, check it thoroughly.
- (3) DAPI staining in high mag should be shown in all the figs with clones as it is difficult to see the suggested changes in the clones at such low mag and w/o knowing whether there are cells and tissue in this focal plane or not (eg. Figure 2A, B, 1F, G, etc)
- (4) Authors claim (based on data in Figure 3) that "these findings collectively suggest that E75 acts genetically in parallel with *Yki*". I am not sure whether the word "in parallel" would describe these data the best. Both proteins appear to be required for each other function, right?

Response to reviewers' comments (#EMBOJ-2024-117040):

Dear reviewers and editor,

Thank you very much for your favorable review and your insightful comments regarding our manuscript. We have conducted additional experiments and made the necessary revisions to address the concerns raised. The following is a point-by-point response to reviewers' comments. For your convenience, we have highlighted the changes made in the revised manuscript using red font. We include most of the new results in the revised manuscript. However, due to space limit, some results and explanations are included only in this rebuttal letter.

We sincerely hope that you will find the revised manuscript to be satisfactory.

Referee #1

This manuscript by Wang and colleagues reports on the role of E75, a nuclear receptor and early ecdysone-responsive gene, in tumor progression. The authors found that the level of ecdysone, an insect hormone, was relatively low in malignant tumors. Using genetic tools, they demonstrated that E75 overexpression increases the malignancy of tumors, while knocking down E75 decreases tumor size. Mechanistically, it was shown that E75 can integrate the Hippo and Notch signaling pathways through physical interaction with Hippo and Notch transcription factors, and by co-binding to their targets. Lastly, the authors demonstrate that the function of E75 in regulating Notch and Hippo pathway-mediated tumorigenesis is conserved in mammals. Overall, this work illuminates the role of E75 in integrating the Hippo and Notch pathways during tumor progression.

Response: Thank you for taking the time to review our manuscript. In the following sections, we will address the specific points that you have raised.

Major concerns:

1. *In the methods section, the authors mentioned that "third-instar larvae imaginal discs" were used. Were the tumors shown in all figures from the same day as the controls? Is the larger tumor size due to a larval developmental delay, resulting in the tumors having extra days to develop? The comparison of tumors in the rescue experiments should also be on the same timeline.*

Response: The tumor figures from the rescue experiments (Figs. 1B, 2D, EV1E, 3E, 4D, and 4E in revision) were dissected on the 6th day after egg laying, same as the controls. The only exception is the tumor with the genotype of *scrib*^{-/-}, *Ras*^{V12} in Fig.1B', as the majority of those tumor-bearing animals developed into giant larvae, which is considered as a typical feature of malignancy in flies. We have unified and described the timeline in the method and indicated in the figure legend panels that were dissected at a different timepoint in the revised manuscript.

2. *Overexpression of E75 could have effects in addition to blocking the EcR signal; it is probably not equivalent to the "hormone inhibition" stated in the abstract. Also, EcR signaling has been shown previously to have a tumor suppressor role in Drosophila (Jiang et al., 2018).*

Response: Thank you for presenting this logical argument. We concur with your viewpoint that overexpression of E75 does not equate to hormone inhibition. This point is supported by the provided evidence. Firstly, we examined the co-expression of several well-established downstream genes of EcR in the *scrib*^{-/-}, *wts*^{-/-} tumors, such as *Hr3*, *ftz-fl*, *Eip93F*, and *br*. Surprisingly, none of these genes could replicate the E75-induced malignant transformation observed in *scrib*^{-/-}, *wts*^{-/-}

tumors (**Author response Figure 1**). These findings demonstrate a diverse and even opposite function of EcR target genes in regulating tumorigenesis, highlighting the distinctive role of E75 in driving tumor malignancy.

Secondly, in line with the conflicting role of EcR in controlling tumor overgrowth, we also observed that inhibiting *EcR* directly using dominant negative forms effectively suppressed tumor overgrowth induced by *scrib*^{-/-}, *wts*^{-/-} (**Author response Figure 1**). Moreover, the overexpression of *EcR* substantially suppressed *Igf*^{-/-}, *Ras*^{V12}-induced tumorigenesis (Figs. EV1E and EV1E'). In combination, these results support our contention that EcR exerts both promoting and suppressing effects on tumor growth in a context-dependent manner.

Author response Figure 1. Eye-antennal discs bearing *ey-Flp-MARCM*-induced mosaics of indicated genotypes. Note that inhibition of *EcR* or overexpression of *Hr3*, *ftz-f1*, *Eip93F*, and *br* significantly suppressed *scrib*^{-/-}, *wts*^{-/-} tumor overgrowth.

To further confirm that E75 overexpression induced overgrowth is indeed dependent on EcR, we co-expressed *EcR* with E75 and found that it rescued the growth phenotype induced by E75 (**Author response Figure 2**). Notably, our findings also demonstrated that ectopic expression of E75 can inhibit endogenous EcR signaling, as evidenced by the downregulation of *EcR* transcription and Br expression (Figs. 2A and 2B). Taken together, these data suggest that E75 overexpression-induced tumor overgrowth is partially due to EcR inhibition.

Author response Figure 2. Eye-antennal discs bearing *ey-Flp-MARCM*-induced mosaics of indicated genotypes. Note that expression of *EcRA* significantly inhibits E75-induced growth.

Regarding the known tumor suppressor role of EcR in the 2018 *Nat. Commun.* paper, instead of using MARCM-induced clonal primary tumors, the authors transplanted *ph*^{-/-}; +*EcR*^{DN} cells into adults with very low ecdysone activity, which might have introduced an artifact. Additionally, our major focus in current manuscript is E75 rather than ecdysone, and we have clearly elucidated the mechanisms through which E75 functions as an oncogene, triggering malignancy. We have made necessary changes in the revised manuscript to reduce potential confusion.

3. The majority of experiments involved E75 overexpression, which could artificially affect *Cut* and *Tag* results.

Response: It seems that you may have overlooked some of our data. We actually investigated the

effect of *E75* depletion by utilizing two RNAi lines (v44851 and BL26717) in different tissues. Consistent with our CUT&Tag data that *E75* could directly bind and regulate the expression of target genes in the Notch and Hippo pathways, our results indicated that inhibition of *E75* significantly reduced Notch signaling activation, as demonstrated by the expression of *E(spl)mβHLH-lacZ* and *Cut* (Figs. 4B, EV3D, and EV3F in revision). Furthermore, depletion of *E75* also downregulated the endogenous expression of multiple target genes in the Hippo pathway (Fig. 3D).

4. In mammalian studies, does inhibition of *NR1D2* lead to hormonal dysregulation? If so, which hormones are affected?

Response: Thank you for raising this question. We reanalyzed the RNA-seq data obtained from U87 cells wherein *NR1D2* knock-down was implemented. We performed the functional annotation of the DEGs and identified a total of 15 hormone-related terms that were enriched, comprising a set of 201 DEGs. Among these DEGs, 69 genes exhibited up-regulation, while 132 genes demonstrated down-regulation. Notably, the knock-down of *NR1D2* significantly affected two major hormone categories: steroid hormones and thyroid hormones (**Author response Figure 3**). These data have been included in the revised manuscript.

Author response Figure 3. Enrichment of hormone related terms between U87 cells and *NR1D2* depleted U87 cells from BP, KEGG, and GO analyses.

5. *NR1D1* is upregulated in GBM, but *E75* is downregulated in fly malignant tumors. This discrepancy should be addressed.

Response: We apologize for any confusion or misunderstanding caused regarding the upregulation of *NR1D2* in GBM. Based on our data, both *E75* and *NR1D2* are considered as oncogenes conceptually. *E75* expression promotes tumor growth, so does *NR1D2*. As an ecdysone-induced gene, the downregulation of *E75* in malignant tumor is a consequence of reduction of ecdysone signaling. Currently, we have limited understanding of why EcR signaling is downregulated in malignant fly tumors. However, it would be an oversimplification to conclude that the expression of every oncogene must be upregulated in tumors. Similarly, multiple target genes of Yki/YAP serve as upstream regulators of the Hippo pathway and should be classified as tumor suppressors. However, their levels are still upregulated when Yki/YAP is activated, which is a common phenomenon observed in various tumors. We revised the manuscript to more accurately and consistently interpret our findings.

6. In the abstract, the authors mentioned that "malignant epithelial tumors in *Drosophila* exhibit a comparable decrease in the activation of hormone signaling." Does this mean malignant tumors exhibit reduced ecdysone signaling? Or does inhibition of ecdysone signaling promote tumor development into malignant ones?

Response: We apologize for the ambiguous expression. This means that malignant tumors exhibit a decrease in ecdysone signaling. We made the necessary modifications to clarify this in the revised manuscript.

7. As shown, knockdown of *E75* reduces tumor formation in *scrib*^{-/-}, *wts*^{-/-}. What about KD of *E75* in *scrib*^{-/-} + *Ras* tumors? Was tumor size reduced?

Response: As suggested, we depleted *E75* in *scrib*^{-/-}, *Ras* tumors, but it did not significantly affect the tumor overgrowth phenotype (Author response Figure 4A). *E75* is one of the primary response-gene of ecdysone signaling, given that ecdysone signaling (represented by *EcRE-lacZ*) is dramatically inhibited in *scrib*^{-/-}, *Ras*^{V12} tumors (Fig. 1F). Therefore, it is not surprising that a further reduction of *E75* had a limited effect on tumor growth. To further explore whether *E75* knockdown can mitigate tumor overgrowth in contexts other than *scrib*^{-/-}, *wts*^{-/-}, we depleted *E75* in another non-malignant tumor induced by the ectopic expression of the Notch intracellular domain (NICD), the activated form of Notch (*N^{act}*). We found that inhibition of *E75* also impeded *N^{act}*-induced overgrowth (Author response Figure 4B).

Author response Figure 4. (A) Dorsal views of *ey-Flp-MARCM*-induced GFP-positive tumor-bearing larvae (left) and the corresponding tumor (right). (B) Representative images of eye-antennal discs bearing *ey-Flp-MARCM*-induced mosaics of indicated genotypes. Quantification of the relative size of GFP positive regions in A and B (A', B').

8. There are three isoforms of *EcR* in the fly, differing only at their N termini. The authors showed two in Fig. S1D. What about the third one?

Response: We co-expressed the third isoform of *EcR* (*EcRC*) and found that it also dramatically suppressed *scrib*^{-/-}, *Ras*^{V12}-induced malignancy (Author response Figure 5). These data have been included in the revised manuscript.

Author response Figure 5. Dorsal views of *ey-Flp-MARCM*-induced GFP-positive tumor-bearing larvae and the corresponding tumor (right). Quantification of larvae pupation rate and relative tumor size are shown.

9. *S2* cells were used for co-IP assays to examine the physical interactions between *E75* and the aforementioned transcription factors (TFs). However, results might differ from those in tumor

tissues.

Response: Thanks for bringing up this concern. It appears that you may have overlooked some crucial data in our study. Our research focused not only on interactions in S2 cells but also within the context of tumors. In original Figure 5F, we conducted co-IP assays using fly tumors that overexpressed Flag-tagged E75, NICD, and Yki, and observed that E75 could physically interact with Yki and NICD. Additionally, we have also validated the physical interactions *in vivo* using the proximity ligation assay (PLA) (Figs. 5A-5C, 5G in original MS). In the revised manuscript, we further examined the physical binding between E75 and Sd or Su(H) in the fly tumor context. As shown in **Author response Figure 6**, both Sd and Su(H) could form a complex with E75 in tumor samples. We have made appropriate modifications to the text and related figure legends to accurately reflect these findings in the revised manuscript.

Author response Figure 6. Co-IP assay to detect physical interactions between E75-Flag and Sd-HA and Su(H). Lysates from *ey-Flp*-MARCM-induced clonal tumors with indicated genotypes were immunoprecipitated and probed with the indicated antibodies.

10. In Fig. 2A, the expression of *EcRE-lacZ* in the E75 OE clone does not show significant changes.

Response: We sincerely apologize for the lack of clarity in the presentation of the figures. As shown in the images below, when examined at a higher magnification, it becomes evident that the expression of endogenous *EcRE-lacZ* is significantly downregulated upon clonal expression of E75, both in the antennal and eye disc regions (**Author response Figure 7**). We have replaced the original images with higher magnification ones in the revised manuscript to address this issue.

Author response Figure 7. Confocal images of eye-antennal discs bearing *ey-Flp*-MARCM-induced mosaics of wild-type and E75 overexpression stained with anti- β -galactosidase antibody for the *EcRE-LacZ* staining.

11. Tumor invasive behavior requires *Mmp1* activity (PMID: 17301221). In Fig. 2E, E75 OE induces *Mmp1* expression in a non-cell autonomous manner? why migrating tumor cells do not show *Mmp1* expression? As *Timp* was upregulated in the E75 overexpression samples (Fig. 3A), is it possible that *Mmp1* expression was inhibited in *scrib*^{-/-}, *wts*^{-/-}+E75 tumors?

Response: Thank you for raising this issue. We conducted a thorough review of our data related to Fig. 2E and other panels. Our analysis indicates that the E75 overexpression upregulates MMP1 upregulation cell-autonomously, as shown in a higher exposure time for GFP channel (**Author**

response Figure 8A). Information present in our initial manuscript stems from the comparatively lower intensity of GFP observed in the protrusion-like structure, particularly when contrasted with the significantly brighter tumor cells. We have replaced the original images in case of further misunderstanding (Author response Figure 8B).

Author response Figure 8. Confocal images of optic lobes from *ey-Flp-MARCM*-induced tumors with indicated genotype stained with anti-Mmp1 antibody.

12. Fig. 3A, are the samples from the same day? Samples in Fig. 3A contain *wts* mutant clones, is the Hippo inactivation in E75 overexpression due to mutant clone expansion of *wts*^{-/-}?

Response: For Fig. 3A, both samples were collected from larvae on the 6th day after egg laying. We believe that the inactivation of Hippo in E75 overexpressed cells is not simply due to mutant clone expansion of *wts*^{-/-}. **Firstly**, Figs. 3B and 3D demonstrate that E75 overexpression alone is sufficient to inactivate Hippo signaling, resulting in the upregulation of Yki target genes. **Secondly**, our CUT&Tag analysis has revealed multiple E75 binding sites located within the promoter regions of canonical Hippo target genes, such as *Diap1*, *ex*, *wg*, and *ft* (Fig. 3I in the revision), suggesting that E75 directly regulates the expression of Hippo signaling target genes. **Thirdly**, ectopic E75 expression synergistically enhanced *scrib*^{-/-}, *wts*^{-/-}-induced upregulation of Hippo target genes, including *Diap1-LacZ* and *ex-LacZ* (Author response Figure 9). We have included these data in the revised manuscript.

Author response Figure 9. Confocal images of eye-antennal discs bearing *ey-Flp-MARCM*-induced tumors with indicated genotype stained with anti-β-galactosidase antibody for the *Diap1-LacZ* and *ex-LacZ* staining.

13. Also in Fig. 3A, since clones were expanded in the *scrib*^{-/-}, *wts*^{-/-}+E75 (Fig. 3d), how come *wts* expression was upregulated in the *scrib*^{-/-}, *wts*^{-/-}+E75 larvae?

Response: The *wts* (*lats*) mutant line was first identified by Tian Xu (Xu *et al.*, 1995,

Development). It is the most used amorphic allele of *wts*, although the nature of the lesion it carries remains unknown. Our hypothesis is that the clones carrying this homozygous allele can initiate the transcription of *wts* mRNA, however, the resulting transcript cannot be translated into a mature and functional protein. In RNA-seq experiments, where sequencing reads typically range from 100 to 300 bp in length, we can still identify the short form of the *wts* transcript and measure its abundance.

Conceptually similarly, we have previously examined the *in vivo* transcriptional changes of *ex* using an *ex* amorphic allele (ex^{e1}) that also carries a *lacZ* reporter insertion (#BL44249). As shown below, we can still detect strong *ex* transcription upregulation in $ex^{-/-}, Ras^{V12}$ clones, as indicated by the *lacZ* staining (Author response Figure 10). This indicates that amorphic allele can still initiate gene transcription.

Author response Figure 10. Confocal images of eye-antennal discs bearing *ey-Flp*-MARCM-induced clones with indicated genotype stained with anti- β -galactosidase antibody for the *ex-LacZ* staining. Note the strong *ex-lacZ* staining in $ex^{-/-}, Ras^{V12}$ clones.

Moreover, upon conducting a more comprehensive examination of *wts* transcription in various tumors, it was found that the expression of *wts* in *scrib*^{-/-}*wts*^{-/-}+E75 tumor is not only elevated in comparison to *scrib*^{-/-}*wts*^{-/-} tumor, but also higher compared to *Ras*, *scrib*^{-/-} tumor. Our CUT&Tag analysis revealed a strong binding of E75 to the *wts* gene (Author response Figure 11), indicating that the direct binding of E75 to *wts* locus may contribute to amplified abundance of *wts* transcripts.

Author response Figure 11. Browser shots of E75 CUT&Tag signal at *wts* gene locus. The shaded areas correspond to E75 peaks.

14. The conclusion that "E75 negatively regulates Hippo signaling" may not be fully supported by the data presented. The statement that 'Consistently, E75 overexpression or depletion could upregulate or downregulate the expression of multiple target genes in the Hippo pathway, including Wingless (*Wg*), expanded (*ex*), Cyclin E (*CycE*), and bantam (*ban*) (Fig. 3C)' may need to be revised based on the specific results shown in Figure 3C. Since E75 overexpression did not change the expression of *Wg* and the expression of *Wg*, *ex*, *CycE*, and *ban* was not affected in the E75 RNAi samples. It is possible that only high levels of E75 are capable of affecting the transcription of Hippo pathway genes.

Response: Thank you for pointing this out, we sincerely apologize for the inadvertent misrepresentation of the data depicted in original Figure 3C (Fig. 3D in the revision). As shown

below, we cropped the same images from our original panels and displayed them at a higher magnification. The data still support our main conclusion that the depletion of *E75* significantly downregulates multiple *Yki* target gene expression (Author response Figure 12), although the phenotype is weaker than that of *E75* overexpression. We have included these data in the revised manuscript. We have removed *Wg* staining panels and included *Diap1-lacZ* staining panels.

Author response Figure 12. Confocal images pouch regions of wing discs with indicated genotype stained with corresponding antibodies. Quantification data is shown on the right.

In addition to *E75* depletion using *ptc*-Gal4, we also employed *ey-flp*-induced MCARM clone, *nub*-Gal4, and *GMR*-Gal4 to deplete *E75* (Fig. 3C, 3F, and S3D in the revision). As shown below, all these methods produced phenotypes that were comparable to those observed during Hippo signaling activation or *yki* knockdown, including apoptosis induction and size reduction (Author response Figure 13).

Author response Figure 13. (A) Confocal images of eye-antennal discs bearing *ey-Flp*-MARCM-induced clones. (B) Wing discs of indicated genotype stained with anti-Dcp-1 antibody. (C) Light micrographs of the adult wings (top) and eyes (bottom) are

Minor comments:

1. Tumor size quantification is required for Fig. 1B.

Response: We quantified the tumor size in the revised manuscript (Author response Figure 14).

Author response Figure 14. Eye-antennal discs bearing *ey-Flp*-MARCM-induced mosaics of indicated genotypes. Quantification of relative GFP⁺ clone size is shown.

2. The expression of *Cut* seems was not reduced in the *E75RNAi* clone. Also, dotted line in fig. S4D *E75RNAi* was misplaced.

Response: I agree with your observation that the change in *Cut* expression upon *E75* depletion may not be extensive. However, I hope you can appreciate that the expression of endogenous *Cut* is indeed decreased in *E75* knockdown clones, compared with surrounding endogenous control cells (**Author response Figure 15A**). Additionally, our findings demonstrate that the depletion of *E75* in the posterior region of the wing disc, under the control of the *hh* promoter, significantly reduces *Cut* expression (**Author response Figure 15B**). Furthermore, staining of *E(spl)mβ-HLH-LacZ* in the wing disc provides further evidence that *E75* is crucial for endogenous Notch signaling activation (**Author response Figure 15C**). These results collectively imply that endogenous *E75* plays a more substantial role in regulating Notch activity in the wing discs.

Author response Figure 15. Eye-antennal discs bearing *ey-Flp*-MARCM-induced mosaics (A) or wing discs (B and C) of indicated genotypes stained with corresponding antibodies.

3. Fig. 4G, *Myc-lacZ* from *E75* overexpression was rotated.

Response: Thank you for pointing this out, we have addressed this issue in the revised manuscript.

4. Samples for DNA binding profiles analysis were from *E75* overexpression? Does *E75* overexpression cause additional binding in the genome?

Response: Thank you for the question. Yes, we overexpressed Flag-tagged *E75* for CUT&Tag analysis, we did try to generate an antibody specific to *E75* to assess the binding profile of the endogenous protein. Unfortunately, the performance of the antibody was unsatisfactory.

5. Mammals do not produce ecdysone. If the malignancy of fly tumors is due to ecdysone or larval developmental delay, how can the authors conclude that the function of E75 in tumor malignant transformation is conserved in mammals?

Response: Thank you for your insightful question. While it is true that mammals do not produce ecdysone, the steroid hormone signaling pathways, including those mediated by nuclear hormone receptors, are highly conserved across species. In our study, we focused on the role of the E75 nuclear receptor, which is a homolog of the mammalian NR1D2. Our data in *Drosophila* and mammals demonstrate that both E75 and NR1D2 function as oncogenes and promote tumorigenesis, therefore, we conclude that the function of E75 in controlling tumor malignant transformation is conserved in mammals. Furthermore, at a molecular level, both E75 and NR1D2 have the ability to integrate Hippo and Notch signaling. We have made necessary modifications to the text in the revised manuscript.

Referee #2

The manuscript of Ma and colleagues unravels a fundamental role of E75 in promoting tumor malignancy in epithelial tumors in Drosophila (and in glial tumors in mammals) by positively interacting (genetically and molecularly) with the Notch and Hippo/Yorkie transcriptional machineries. Based on the role of E75 in repressing ecdysone signaling and on the fact that ecdysone signaling is repressed in malignant tumors, authors propose that their experimental model is a paradigm of tumor malignancy caused by hormone depletion. The MS combines the use of Drosophila epithelia and a mammalian glial model to demonstrate conservation. The MS is well written, figures self-explanatory and topic very timely.

Response: Thank you for recognizing the overall significance and the relevance of our research topic. We have made necessary revisions to further improve the manuscript.

Major points:

1. Authors claim that the effects of E75 on tumor growth are caused by changes in ecdysone signaling. Authors should validate some of their results (cooperative oncogenesis with *scrib wts*, or others) by depleting ecdysone signaling with other tools.

Response: Thanks for raising this important concern. Our original data suggest that ecdysone signaling is inhibited in malignant tumors and we found that ectopic overexpression of EcR can rescue the tumor overgrowth phenotype. Following your suggestion, we performed a comprehensive analysis and revealed a complex interaction between EcR and tumorigenesis. We inhibited EcR directly using dominant negative forms and observed an effective suppression of tumor overgrowth induced by *scrib^{-/-}*, *wts^{-/-}* (**Author response Figure 16**). This implies that a strong inhibition of ecdysone signaling prevents tumor malignancy.

We hypothesize that a robust inhibition of ecdysone signaling can yield multifaceted effects on cell proliferation, owing to the broad inhibition of multiple downstream target genes of EcR. Notably, a recent study by Jean-Paul Vincent's group revealed that the ecdysone signal can exert contradictory influences on proliferation. Depending on the systemic concentration of the ligand level, the ecdysone receptor has the capacity to either promote or suppress proliferation (Gantas Perez-Mockus *et al.*, 2023, *Developmental Cell*, PMID: 37769663). To further examine the tumor regulating role of EcR target genes, we overexpressed several well-established downstream genes of EcR in the *scrib^{-/-}*, *wts^{-/-}* tumors, such as *Hr3*, *ftz-f1*, *Eip93F*, and *br*. Surprisingly,

overexpression of these genes inhibits tumorigenesis, none of them could phenocopy E75-induced malignant transformation observed in *scrib*^{-/-}, *wts*^{-/-} tumors (Author response Figure 16). These findings demonstrate a diverse and even opposite function of EcR target genes in regulating tumorigenesis, highlighting the distinctive role of E75 in driving tumor malignancy.

Author response Figure 16. Eye-antennal discs bearing *ey-Flp*-MARCM-induced mosaics of indicated genotypes. Note that inhibition of *EcR* or overexpression of *Hr3*, *ftz-f1*, *Eip93F*, and *br* significantly suppressed *scrib*^{-/-}, *wts*^{-/-} tumor overgrowth.

Collectively, our data suggest that the regulation of ecdysone signaling on tumorigenesis is complex and requires further investigation. However, the main focus of this manuscript is on E75 rather than EcR. We have presented sufficient evidence to demonstrate that E75 can downregulate EcR activity, and we further confirmed that E75 overexpression-induced overgrowth is indeed dependent on EcR, as co-expression of *EcR* with *E75* rescued the growth phenotype induced by E75 (Author response Figure 17). Additionally, we have shown that E75 drives malignant tumor transformation by integrating Hippo and Notch pathway at the transcription factor level. Although we acknowledge that investigating the underlying mechanisms of EcR in tumor formation would be interesting, it will be beyond the scope of the current manuscript.

Author response Figure 17. Eye-antennal discs bearing *ey-Flp*-MARCM-induced mosaics of indicated genotypes. Note that expression of *EcRA* significantly inhibits E75-induced

2. I wonder whether the effects of *N^{dn}* in repressing tumor growth is just a consequence of the deleterious effects of *N^{dn}* on growth. Some controls would be required. Are *Ndn* expressing clones, cells or tissues affected in growth and survival? I would guess the same happens to *Yki*-RNAi clones where authors might want to use *scalloped*-RNAi tools to circumvent the effect of *Yki* in "normal" development. Similar comment on *dMyc*. Some controls would be required.

Response: As suggested, we added several controls to the revised manuscript. It is worth noting that compared with controls, clonal inhibition of Notch alone in eye-antennal discs by overexpressing a dominant negative form did not significantly affect clone size (Author response Figure 18).

Author response Figure 18. Dorsal views of *ey-Flp-MARCM*-induced GFP-positive clone-bearing larvae and the corresponding eye-antennal discs. Quantification of relative GFP⁺ area size is shown.

We also depleted *sd* in E75 overexpressed clones and observed a dramatic suppression on the upregulation of *Diap1-lacZ* (Author response Figure 19A). Moreover, depletion of *sd* significantly impeded E75+*scrib*^{-/-}-induced tumor overgrowth (Author response Figure 19B).

Author response Figure 19. (A) Eye-antennal discs bearing *ey-Flp-MARCM*-induced mosaics of indicated genotypes are stained with anti-β-galactosidase antibody for the *Diap1-LacZ* staining. (B) Dorsal views of *ey-Flp-MARCM*-induced GFP-positive tumor-bearing larvae and the corresponding eye-antennal discs (right). Quantification of relative GFP⁺ area size is shown.

Similarly, we included *Myc*^{RNAi} controls in the revised manuscript (Author response Figure 20).

Author response Figure 20. (A) Representative confocal images of wing disc bearing indicated genotypes. (B) Eye-antennal discs bearing *ey-Flp-MARCM*-induced mosaics of indicated genotypes.

3. Difficult to understand that the ability of E75 to induce malignancy of *wts*^{-/-}*scrib*^{-/-} tumors relies on its capacity to block hippo signaling, since *hpo* is already depleted in *wts*^{-/-} clones. Don't these data indicate that E75 has some other activities besides its role in Hippo signaling. Indeed, Hippo depletion does not lead to malignancy by its own!!!

Response: We apologize for any confusion or misunderstanding we might cause in our initial manuscript. Indeed, you are correct that E75 possesses functional roles beyond its interaction with

the Hippo signaling pathway. As you astutely observed, the depletion of Hippo pathway activity alone does not suffice to induce malignancy, suggesting the involvement of additional pathways. This is investigated in our manuscript under the section "E75 activates the Notch pathway to promote tumor malignancy." We highlight that the sole activation of Yki is insufficient to account for the full spectrum of malignancy observed in *scrib*^{-/-}, *wts*^{-/-} tumors, indicating that E75's oncogenic effects extend beyond Hippo signaling modulation. We further dissected the underlying mechanisms in Figures 4 and S4 and demonstrate how E75 also engages the Notch signaling pathway activation, contributing to tumor progression. We have revised the manuscript to emphasize this point more clearly and to improve overall readability.

Other points:

(1) Changes in the expression levels of Yki targets in Figure 3C are not clear. First, *Wg* is not a bona fide Yki target. Second, changes are unconvincing in the case of E75 depletion. Some quantification might help.

Response: We apologize for the lack of clarity in the presentation of the data depicted in Figure 3C (Fig. 3D in the revision). To enhance visibility, we have cropped the images and highlighted regions of interest with white arrows. We replaced *Wg* staining-related panels with *Diap1-lacZ* staining images and quantified the fluorescent staining intensity (**Author response Figure 21**).

Author response Figure 21. Confocal images pouch regions of wing discs with indicated genotype stained with corresponding antibodies. Quantification panels are shown on the right.

(2) Grammar/writing issues should be corrected throughout the ms (some words are lacking, etc). Please, check it thoroughly.

Response: We have doublechecked the revised manuscript carefully.

(3) DAPI staining in high mag should be shown in all the figs with clones as it is difficult to see the suggested changes in the clones at such low mag and w/o knowing whether there are cells and tissue in this focal plane or not (eg. Figure 2A, B, 1F, G, etc)

Response: As suggested, we have included DAPI staining for corresponding figures.

(4) Authors claim (based on data in Figure 3) that "these findings collectively suggest that E75 acts genetically in parallel with Yki". I am not sure whether the word "in parallel" would describe

these data the best. Both proteins appear to be required for each other function, right?

Response: Thank you for raising this concern. We believe that the mutual genetic dependence of Yki and E75 for their respective functions implies that neither protein functions upstream or downstream over the other. This suggests that E75 and Yki function in parallel with one another. Supporting this notion, our co-IP findings indicate a physical interaction between E75 and Yki.

Dear Xianjue,

Thank you for submitting your revised manuscript for consideration by The EMBO Journal. I sincerely apologise for the protracted assessment process due to delays in reviewer report submission and extensive consultations within the team. I have now received comments from both original reviewers, which are included below for your information.

Unfortunately, both reviewers find that some of their points were not addressed satisfactorily. Reviewer #1 finds that the cell-autonomous nature of MMP1 upregulation and Yki target downregulation remains insufficiently conclusive and needs further quantification. Furthermore, reviewer #2 finds that the interrelationships between E75 and EcR, Yki and Myc remain unclear and needs further exploration, also in the connection with tumorigenesis. While I appreciate the invested effort in addressing these points, it appears that further investigation would be needed to provide sufficient clarity and conclusiveness that we would need for publication of the manuscript in The EMBO Journal. However, if you find that you can address these remaining points with new data and adjustments of the conclusions as needed, I would be happy to reconsider the decision.

Alternatively, in the interest of a rapid publication of the study, I have discussed your manuscript and referee comments with my colleague Achim Breiling at our sister journal EMBO Reports. I am glad to say that he would be interested to proceed with a significantly reframed version of the manuscript, either strengthening claims on providing insight how hormone inhibition promotes tumor malignancy and links to Hippo and Notch pathways (as indicated in the second round referee reports), or toning down such claims or links. If you are interested in this option, please contact Achim Breiling at a.breiling@emboreports.org with a revision plan.

Thank you in any case for the opportunity to consider this manuscript. I am sincerely sorry that I could not communicate more positive news, and I very much hope that you will find the transfer of interest.

With kind regards,

Ieva

Referee #1:

I appreciate the efforts made by the authors to address most of my earlier concerns, which are mostly satisfactory. However, for comments 11 and 14, the cell-autonomous activation of MMP1 (author response Figure 8) and downregulation of Yki targets (author response Figure 12) are still not obvious from the figures provided. Specifically, for Diap1-lacZ and ex-lacZ, did the authors only quantify a small region of GFP-positive cells or the entire Ptc-Gal4 domain (author response Figure 12)? In Figure 8, there are GFP-negative regions showing MMP1 upregulation, which may suggest non-autonomous JNK activation.

Referee #2:

I appreciate the response of the authors to my major concerns. However, I still have the following issues that need to be addressed by the authors.

(1) Significance of ecdysone signalling downregulation. Authors found out that ecdysone signalling is downregulated in malignant tumors and propose that this downregulation is fundamental for tumor malignancy. This claim is supported by the fact that increasing ecdysone signalling by overexpression of EcR rescues tumor growth and that overexpression of E75 (a target of Ecdysone that transcriptionally represses ECR target) increases tumor malignancy. Fundamental controls are required in this case: Are ecdysone targets (Broad or the EcR responsive elements) restored in tumor cells upon overexpression of the EcR (these data are not provided)? Is EcR overexpression affecting cell viability or clonal growth (no control is shown)? The fact that authors are not able to reproduce the E75 overexpression phenotype upon depletion of EcR suggests that the central element is E75 and not EcR signalling in general. Last question: have authors found E75 upregulation in tumors?

(2) dMyc: the fact that dMyc depletion affects clonal growth in controls (author response 20) invalidates the whole section: "Myc is an essential downstream gene for E75-induced tumor malignancy".

(3) Yki and E75: if there is genetic interdependence, author should not conclude that they are acting in parallel. I am not convinced by their arguments.

** As a service to authors, EMBO Press provides authors with the possibility to transfer a manuscript that one journal cannot offer to publish to another EMBO publication or the open access journal Life Science Alliance launched in partnership between EMBO Press, Rockefeller University Press and Cold Spring Harbor Laboratory Press. The full manuscript and if applicable, reviewers' reports, are automatically sent to the receiving journal to allow for fast handling and a prompt decision on your manuscript. For more details of this service, and to transfer your manuscript please click on Link Not Available. **

Point by point response to reviewers' comments (#EMBOJ-2024-117040R):

Dear reviewers and editor,

Thank you for taking the time to review our revised manuscript and we would like to express our gratitude again to the reviewers for their insightful comments, which have greatly contributed to the enhancement of our manuscript. The revised version of this manuscript now incorporates additional data to address reviewers' concerns.

We sincerely hope that you will find the revised manuscript to be satisfactory.

Referee #1:

I appreciate the efforts made by the authors to address most of my earlier concerns, which are mostly satisfactory. However, for comments 11 and 14, the cell-autonomous activation of MMP1 (author response Figure 8) and downregulation of Yki targets (author response Figure 12) are still not obvious from the figures provided. In Figure 8, there are GFP-negative regions showing MMP1 upregulation, which may suggest non-autonomous JNK activation.

Response: We have conducted additional experiments to address these concerns and apologize for any confusion regarding the Mmp1 staining. As illustrated in the original Figure 8B in the previous response letter, the residual Mmp1 staining signal, which appears to be non-autonomous Mmp1 expression (A, bottom panels), actually indicates the endogenous expression of Mmp1 in the NVC.

(Former author response Figure 8)

To provide further evidence, we conducted antibody staining against Mmp1 and p-JNK (to assess the endogenous expression of phosphorylated JNK) in eye disc clones that overexpressed E75, as well as in tumor-bearing $E75+scrib^{-/-}, wts^{-/-}$ samples. Results from **Author response Figure 1A and B** illustrate a specific upregulation of both p-JNK and MMP-1 in the GFP-positive region. Moreover, we observed an independent upregulation of p-JNK and Mmp1 in the GFP-positive region of the optic lobes in $E75+scrib^{-/-}, wts^{-/-}$ samples, as shown in Author response Figure 2. Collectively, these data indicate that E75 expression induce JNK activation in a cell-autonomous manner. We have included these data in the revised manuscript.

Author response Figure 1. Confocal images of from *ey-Flp*-MARCM-induced eye discs (A) and tumors (B) with indicated genotype stained with anti-Mmp1 and p-JNK antibody.

Author response Figure 2. Confocal images of optic lobes from *ey-Flp*-MARCM-induced eye discs and tumors with indicated genotype stained with anti-Mmp1 and p-JNK antibody.

Specifically, for Diap1-lacZ and ex-lacZ, did the authors only quantify a small region of GFP-positive cells or the entire Ptc-Gal4 domain (author response Figure 12)?

Response: To visualize the changes in Yorkie (Yki) target expression more effectively, we utilized *en-Gal4* or *hh-Gal4* drivers to repeat the antibody staining experiments. The staining on the anterior parts of the wing disc can be considered as an endogenous control. This approach allows for a clearer observation and analysis of Yki targets following the depletion of *E75* specifically in the posterior region of the wing disc. As depicted in **Author response Figure 3**, the expression of multiple Yki target genes in the GFP-positive regions showed a significant downregulation. We have included these data in the revised manuscript.

Author response Figure 3. Confocal images and quantification of wing discs from *en*-Gal4 or *hh*-Gal4 stained with β -gal and CycE antibody.

Referee #2:

(1) *Significance of ecdysone signaling downregulation.* Authors found out that ecdysone signalling is downregulated in malignant tumors and propose that this downregulation is fundamental for tumor malignancy. This claim is supported by the fact that increasing ecdysone signaling by overexpression of *EcR* rescues tumor growth and that overexpression of *E75* (a target of Ecdysone that transcriptionally represses *EcR* target) increases tumor malignancy.

Response: Thank you for bringing up this important concern. We completely agree with you that depletion of *EcR* does not equate to *E75* overexpression. *EcR* plays a crucial role in inducing various downstream target genes, including both tumor repressors and oncogenes, upon binding with ecdysone (see **Author Response Figure 4**). As we mentioned earlier, depletion of *EcR* would repress the expression of all these target genes. However, overexpression of *E75* has a different effect. It not only represses ecdysone signaling, but also represses Hippo signaling and activates Notch signaling. Following your suggestion, we have removed the statement regarding the overexpression of *E75* mimicking ecdysone inhibition to avoid any confusion.

Author response Figure 4. Figure 4 depicts a schematic diagram illustrating the process of ecdysone signaling, the involvement of ecdysone response genes, and its relationship with tumorigenesis.

*Fundamental controls are required in this case: Are ecdysone targets (Broad or the *EcR* responsive elements) restored in tumor cells upon overexpression of the *EcR* (these data are not provided)?*

Response: Thank you for bringing this to our attention. We stained *Br* in *Igt^{-/-}/Ras^{v12}* tumors that co-expressed various isoforms of *EcR*. As depicted in **Author response Figure 5**, the overexpression of all *EcR* isoforms effectively reversed the downregulation of *Br* expression induced by *Igt^{-/-}, Ras^{v12}*.

Author response Figure 5. Confocal images and quantification of Br staining in *ey-Flp-MARCM*-induced eye discs and tumors with indicated genotype

Is *EcR* overexpression affecting cell viability or clonal growth (no control is shown)?

Response: As suggested, we examined clone size changes upon the overexpression of different *EcR* isoforms. As shown in **Author Response Figure 6**, *EcR* overexpression alone did not affect clone size under physically conditions. We have included these data in the revised manuscript.

Author response Figure 6. Confocal images and quantification of *ey-Flp-MARCM*-induced eye discs and tumors with indicated genotype

The fact that authors are not able to reproduce the *E75* overexpression phenotype upon depletion of *EcR* suggests that the central element is *E75* and not *EcR* signalling in general.

Response: We completely agree with you that the central focus of our manuscript is *E75*, as stated in the title. *E75* is an ecdysone-induced gene that serves multiple functions. It acts as a repressor of ecdysone signaling through a negative feedback loop, and it also functions as a regulator of the Hippo and Notch signaling pathways. In the revised manuscript, we have made necessary modifications to minimize potential misunderstandings.

Last question: have authors found *E75* upregulation in tumors?

Response: *E75* is one of the primary response-gene of ecdysone signaling. As the ecdysone signaling (represented by *EcRE-lacZ*) is inhibited in *scrib*^{-/-}, *wts*^{-/-} and *scrib*^{-/-}, *Ras*^{V12} tumors in different degrees (Figs. 1E and F), therefore, it is predictable that *E75* is also downregulated as a consequent of ecdysone signaling reduction.

(2) *dMyc*: the fact that *dMyc* depletion affects clonal growth in controls (author response 20) invalidates the whole section: "*Myc* is an essential downstream gene for *E75*-induced tumor malignancy".

Response: As *Myc* is required for cell proliferation, its downregulation reasonably affects clonal growth. Our data clearly show that both the transcription and protein levels of *Myc* were upregulated upon the overexpression of *E75* (Figs. 4G, 4H, and EV3I). Our genetic data further support the idea that *Myc* mediates *E75*-induced tumor malignancy (Fig. 4I). Furthermore, our CUT&Tag data reveal that *E75* has the ability to bind to the regulatory region of *Myc* (Fig. EV3J). Consequently, we conclude that *Myc* is a direct downstream target gene of *E75*. Given this, it is reasonable to infer

that Myc is necessary for E75-induced malignant tumor transformation. In response to your suggestion, we have modified the section title as follows: "Myc is necessary for E75-induced malignant tumor transformation," and have adjusted the conclusion of that section to read: "These data collectively demonstrate that Myc functions as a downstream target gene of E75, influencing both tumor growth and malignancy."

(3) Yki and E75: if there is genetic interdependence, author should not conclude that they are acting in parallel. I am not convinced by their arguments.

Response: Thanks for bring up this issue, we have deleted "in parallel" in that sentence and modified it as follow: "These findings collectively suggest that E75 and Yki act genetically interdependent with each other."

Dear Xianjue,

Thank you for submitting a revised version of your manuscript. I sincerely apologise for the protracted assessment process due to my absence from the office at the beginning of the September and the unusually high submission rate to our office at the moment.

I have now gone through your revision, and I find that most of the remaining concerns have been addressed satisfactorily. However, in response to the comment by referee #2, since you have performed transcriptome analysis in wing disc tumours, please provide data from this dataset on whether you see changed expression of E75 in these tissues.

Additionally, there are a few editorial points that need addressing before I can extend official acceptance of the manuscript:

1. Please make sure that the funding information is correct and identical both in the manuscript and our online system.
2. Please make sure that the manuscript section order is as follows: title page with complete author information, abstract, keywords, introduction, results, discussion, methods, data availability section, acknowledgements, disclosure and competing interests statement, references, main figure legends, tables, expanded figure legends.
3. CRedit has replaced the traditional author contributions section because it offers a systematic, machine-readable author contributions format that allows for more effective research assessment. Please remove the Authors Contributions from the manuscript and use the free text boxes beneath each contributing author's name in our online submission system to add specific details on the author's contribution. More information is available in our guide to authors.
4. For Dataset EV1, please add a legend in the manuscript file.
5. We noted a potential image reuse between figure panels 2B and EV2A, scrib-/wts- sample. Please check and note this in the figure legend.
6. In our standard source data check, we have noted unexplained numerical duplications in the source data for figure 6H. I have attached the corresponding file with the detected duplications labelled in colour. Please take a look and correct as needed. A brief explanation would be very helpful.
7. Our data editors have flagged the following issues in figure legends that need correcting:
 - Please define the annotated p values **** as well as provide the exact p-values for the same in the legend of figure EV 4d; as appropriate.
 - Please provide the exact p values in the legends of figures 1f', g'; 2d'; 3c', e', f'; 4d', e', i"; 6a-d, h, j; EV 1a, d', e", f', g', h"; EV 2g, h', k'; EV 3h',k', l'; EV 5d-e, g, i.
 - Please indicate the statistical test used for data analysis in the legends of figures 1c-d; EV 2c, e, l; EV 3a; EV 4d; EV 5a-b.
 - Please provide information on the number and nature of replicates in the legends of figures 6h; EV 2g; EV 5d-e, g.
 - Please describe the nature of replicates (e.g., biological or technical) in the legends of figure EV 4d.
 - Please note define the error bars in the legends of figures EV 4d.
 - Please note that the scale bar needs to be defined for figure EV 2k.
 - Please note that scale bar and its definition are missing for figures 1b'; EV 2a.

With best wishes,

leva

leva Gailite, PhD
Senior Scientific Editor
The EMBO Journal
Meyerhofstrasse 1
D-69117 Heidelberg
Tel: +4962218891309
i.gailite@embjournal.org

We realize that it is difficult to revise to a specific deadline. In the interest of protecting the conceptual advance provided by the work, we recommend a revision within 3 months (19th Dec 2024). Please discuss the revision progress ahead of this time with

the editor if you require more time to complete the revisions.

Point by point response to editor's comments (#EMBOJ-2024-117040R2):

1. Since you have performed transcriptome analysis in wing disc tumours, please provide data from this dataset on whether you see changed expression of *E75* in these tissues.

Author response Figure 1. Volcano plot of differentially expressed genes (DEGs) in *scrib*^{-/-}, *wts*^{-/-} and *scrib*^{-/-}, *Ras*^{V12} tumors (above), *E75* is marked in blue. Log₂ Fold Change of *E75* in *scrib*^{-/-}, *wts*^{-/-} tumors. *scrib*^{-/-}, *Ras*^{V12} tumors vs control (bottom).

Figure 1E

Response: Thank you for pointing this out. We have marked *E75* among the differentially expressed genes (DEGs) in *scrib*^{-/-}, *wts*^{-/-} and *scrib*^{-/-}, *Ras*^{V12} tumors in **Figure EV1 B**. Additionally, we present Log₂ Fold Change of *E75* beneath the volcano plot (**Author response Figure 1**). The change in *E75* expression is also reflected in the fifth column from the right in the heatmap profiles of ecdysone signaling in the original **Figure 1E**. Due to the inhibition of ecdysone signaling in the *scrib*^{-/-}, *Ras*^{V12} tumor, *E75* is downregulated in this context.

Dear Xianjue,

Thank you for addressing the final points. I sincerely apologise for the delay in communicating the decision due to the high number of submissions we receive at the moment. I am now pleased to inform you that your manuscript has been accepted for publication.

Before we forward your manuscript to our publishers, I would like to propose some edits in the manuscript title, abstract and synopsis (please see below and the attached manuscript text file). I have also written a short blurb that will accompany the title of your manuscript in our online system. Please let me know if any corrections or adjustments are needed.

Title:

Nuclear receptor E75/NR1D2 promotes tumor malignant transformation by integrating Hippo and Notch pathways

Blurb:

The conserved steroid-hormone-induced transcription factor E75/NR1D2 interacts with Hippo and Notch pathway transcription factors to co-regulate their activity.

Synopsis:

The interplay between hormone signaling and malignant transformation remains incompletely understood. This study identifies E75/NR1D2 as an evolutionarily conserved oncogene that promotes malignant transformation of benign tumors by co-regulating the Hippo and Notch signaling pathways.

- The ecdysone steroid hormone signaling is inhibited in malignant *Drosophila* tumors.
- Overexpression of nuclear receptor E75 facilitates malignant tumor transformation.
- E75 promotes tumor malignancy by integrating Hippo pathway and Notch pathway signaling at the transcription factor level.
- The mammalian E75 ortholog also regulates the Hippo and Notch pathways and promotes glioblastoma progression.

Finally, we would like to promote your manuscript among the Chinese readership. Therefore, we would like to invite you to prepare a short summary of the manuscript in Chinese (1500-2000 Chinese characters), which we will promote on the WeChat platform 'BioArt' with more than 610,000 followers.

If you are interested in this opportunity, we recommend covering the article very close to its online publication date. Thus, ideally we would very much appreciate if you could send us a draft within the next 7 working days. Please let us know whether or not you would be interested in contributing such a short summary in Chinese.

I have included below some general guidelines on how to prepare a summary and a link to recent examples for your reference. Please let me know if you have any questions about this.

If you have any questions, please do not hesitate to contact the Editorial Office. Thank you for this contribution to The EMBO Journal and congratulations on a nice study!

With best wishes,

Ieva

Ieva Gailite, PhD
Senior Scientific Editor
The EMBO Journal

Meyerhofstrasse 1
D-69117 Heidelberg
Tel: +4962218891309
i.gailite@embojournal.org

General WeChat Summary Guidelines

1. These summary articles are meant to be targeting general audience so please limit the use of specialized technical terms, acronyms and jargon.
2. A summary usually starts with brief background information of the reported work, which is followed by explaining the findings in some detail, and ends with a short review of the conclusions as well as the implications of the work and future directions for the research.
3. The summary should at least contain one graphical item, such as a scheme or a figure from the paper.
4. Please provide ONE SINGLE document containing all text and graphical materials, ideally as a Word.docx or .doc file. Please DO NOT provide the document as a .pdf file.
5. Please DO NOT publicly release the document before the paper is officially published online.

Summary Examples

EMBO J | 罗招庆/欧阳松应揭示谷酰胺脱氨酶MvcA的去泛素化功能

EMBO J | 王松灵院士团队揭示组织内应力调控大型哺乳动物乳恒牙替换的新机制
